# Balancing Perception and Distortion in Super Resolution via Spatial-Semantic Guidance

## Abstract

In image super-resolution (SR), perceptual quality and distortion form two competing objectives, bounded by the Perception-Distortion trade-off. GAN-based SR models reduce distortion but often fail to synthesize realistic fine-grained textures, while diffusion-based models generate perceptually plausible details but frequently hallucinate content, leading to fidelity loss. This raises a key challenge: how to harness the powerful generative priors of diffusion models without sacrificing fidelity. We introduce **SpaSemSR**, a Spatial-Semantic guided diffusion-based framework that addresses this challenge through two complementary guidance. First, **spatial-grounded textual guidance** integrates object-level spatial cues with semantic prompts, reducing distortion by aligning textual guidance with visual structure. Second, **semantic-enhanced visual guidance** unifies multimodal semantic priors via a multi-encoder design with semantic degradation constraints, improving perceptual realism under severe degradations. These complementary guidances are adaptively fused with diffusion priors via novel spatial-semantic attention mechanisms, curbing distortion and hallucination while preserving the strengths of generative diffusion models. Extensive experiments across multiple benchmarks demonstrate that SpaSemSR achieves a state-of-the-art balance between perception and distortion, producing both realistic and faithful restorations.

## 1 Introduction

Image Super-Resolution (SR) aims to reconstruct high-resolution (HR) images from low-resolution (LR) inputs degraded by complex and often unknown processes, with the dual objective of achieving high fidelity and strong perceptual quality. However, as demonstrated by (Blau & Michaeli, 2018), SR methods are fundamentally constrained by the **Perception-Distortion trade-off**: improving distortion (i.e., reconstruction accuracy) inevitably comes at the cost of perceptual quality, and vice versa, due to the monotone boundary between the two. Distortion is measured by full-reference metrics such as PSNR and SSIM, while perception emphasizes visual realism regardless of ground-truth similarity, and is assessed by reference-free metrics such as CLIP-IQA, MUSIQ, and MANIQA.

GAN-based methods (Liang et al., 2022; Zhang et al., 2022; 2021; Wang et al., 2021) have demonstrated strong fidelity performance, but their gains in reconstruction accuracy do not always translate into perceptual improvements (e.g., lower CLIP-IQA, MUSIQ). As illustrated in Fig. 1, these models often produce artifacts or blurring issues, stemming from unstable adversarial training, domain shifts between synthetic training and real-world test data, and fidelity-biased optimization objectives. Recently, diffusion-based approaches (Wang et al., 2024; Lin et al., 2024; Yang et al., 2024; Wu et al., 2024c; Qu et al., 2024; Chen et al., 2025) have emerged as powerful alternatives. Leveraging the rich generative priors of large-scale text-to-image (T2I) diffusion models, they excel in generating realistic textures and achieve superior scores on perceptual quality metrics. Yet, somewhat counter-intuitively, these methods often underperform GAN-based models on distortion metrics (e.g., lower PSNR and SSIM), leading to reduced fidelity and frequent hallucinations, as shown in Fig. 1. More recently, lots of studies (Wu et al., 2024b; Dong et al., 2025; Zhang et al., 2025) have shifted their focus toward improving the efficiency of diffusion models, aiming to design more effective one-step generation schemes for Stable Diffusion, thereby reducing the heavy computational cost caused by the multi-step denoising process. However, these approaches do not primarily focus on addressing the distortion problem caused by hallucinatory artifacts generated in diffusion models. Thus, neither paradigm fully resolves the tension between perception and distortion, leaving it as an unsolved issue.

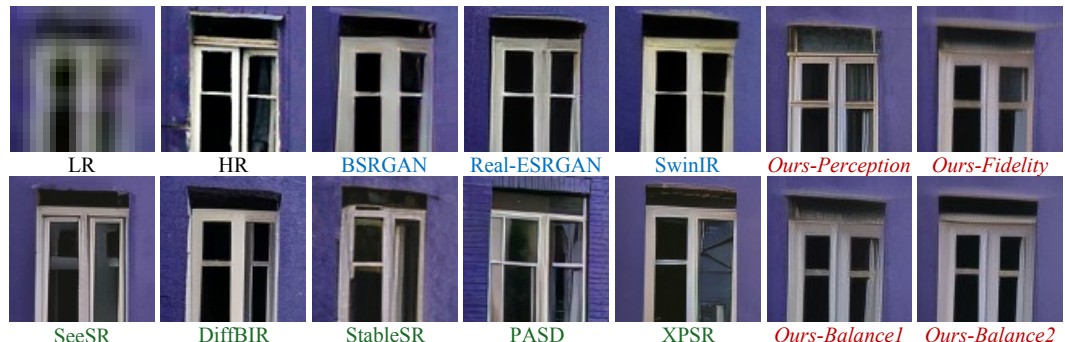

Figure 1: Perception-distortion trade-off in GAN-based, Diffusion-based, and *Our four variants* (Sec. 3.1): GAN-based methods yield less distortion but blurry textures, diffusion-based methods produce sharp yet hallucinatory details, while our four variants restore clean, semantically accurate textures with high fidelity.

To further investigate the Perception-Distortion trade-off, we revisit the mechanisms underlying diffusion-based SR. Despite their strong generative priors, diffusion models face three key challenges: (1) Their generative nature promotes sample diversity, which benefits image synthesis but undermines fidelity in restoration. (2) Severe degradations in LR inputs often destroy local structures, leading to ambiguous semantics. (3) Existing solutions, such as PASD (Yang et al., 2024), SeeSR (Wu et al., 2024c), address this by introducing semantic text prompts as auxiliary conditions. However, text prompts alone lack spatial awareness, and thus reconstructions remain distorted compared to the ground truth, yielding high perceptual but low fidelity scores (Ren et al., 2025).

In this work, we present a **Spa**tial-**Sem**antic guided SR framework (**SpaSemSR**), a diffusion-based approach that pushes the perception-distortion boundary by integrating complementary spatial and semantic guidance. Specifically, we introduce: a. **spatial-grounded textual guidance**, which aligns spatial information with semantic textual prompts to improve fidelity. b. **semantic-enhanced visual guidance**, which enriches perceptual quality by constraining generation with multimodal semantic priors. Together, two forms of guidance bridge the gap between semantically rich but spatially ambiguous text prompts and spatially precise but semantically degraded visual features.

To achieve it, **firstly**, we introduce a novel spatial-aware text fusion mechanism for better fidelity representation that integrates object-level spatial coordinates with corresponding semantic textual tags. **Secondly**, to extract rich and robust semantics from degraded LR inputs, we design a two-branch image encoder system: one captures low-level latent structures, while the other extracts high-level semantic features. The perception ability of these extracted features is further ensured by our novel semantic degradation constraints derived from pre-trained VAE (Rombach et al., 2022) and SAM (Kirillov et al., 2023) encoders. **Thirdly**, to effectively integrate semantic and spatial priors with diffusion generative priors, we propose our Spatial-Semantic ControlNet, Spatial-aware Text Attention (SpaTextAtten), and Semantic-enhanced Image Attention (SemImgAtten) layers, enabling adaptive fusion of spatial and semantic guidance across modalities in the diffusion model. As a result, this spatial and semantic guidance yields reconstructions that are both perceptually compelling and faithful to the original content, thereby achieving a balance between perception and fidelity. Our contributions are summarized as follows:

- We propose SpaSemSR, a novel diffusion-based framework for balancing perception and distortion, which jointly exploits complementary spatial and semantic guidance, curbing distortion and hallucination while fully leveraging the strengths of diffusion priors.

- We introduce a spatial-aware text fusion mechanism that augments semantic prompts with spatial grounding, thereby improving generation fidelity and alleviating spatial misalignment between textual and visual representations.

- We design a semantic-enhanced multi-encoder architecture with semantic degradation constraints that jointly capture low-level structures and high-level semantics, further constrained by pretrained VAE and SAM priors to provide robust perceptual learning.

- We propose the Spatial-Semantic ControlNet, SpaTextAtten, and SemImgAtten layers to effectively integrate semantic and spatial guidance into diffusion-based generation.

- Extensive experiments demonstrate that SpaSemSR substantially reduces blurry textures and hallucinatory artifacts, achieving a state-of-the-art balance between perception and

distortion across multiple benchmarks. Ablation studies further validate the complementary contributions of spatial and semantic guidance to fidelity and perceptual quality.

## 2 RELATED WORK

**Image Super-Resolution.**    Classical SR methods (Gu et al., 2019a; Huang et al., 2020; Zhang et al., 2018) estimate predefined degradation kernels to recover high-resolution images. While effective on synthetic data, they struggle with complex real-world degradations. To address this, BSRGAN (Zhang et al., 2021) introduced randomized degradation pipelines, and Real-ESRGAN (Wang et al., 2021) proposed high-order degradations with Sinc filters. GAN-based SR further incorporated perceptual losses to enhance visual quality, but often introduces artifacts and fails to reconstruct faithful textures, motivating the use of stronger generative priors.

**Diffusion Prior-based Super-Resolution.**    Diffusion models leverage powerful generative priors to produce perceptually realistic SR results. StableSR (Wang et al., 2024) fine-tuned Stable Diffusion with a Time-aware Encoder and controllable feature wrapping. DiffBIR (Lin et al., 2024) used restoration modules and IRControlNet to remove degradations while preserving fidelity. Text-guided SR methods PASD (Yang et al., 2024), SeeSR (Wu et al., 2024c), FaithDiff (Chen et al., 2025), XPSR (Qu et al., 2024)) extract semantic cues from images or multimodal models to guide generation. However, these approaches often use global or loosely aligned text prompts, lacking precise spatial grounding. Our work explicitly integrates spatial-aware semantic guidance into diffusion for fidelity restoration.

**Perception-Distortion Trade-off.**    Balancing perceptual realism and fidelity is a key SR challenge. The perception–distortion trade-off was formalized in (Blau & Michaeli, 2018). Subsequent works addressed it via multi-objective strategies: a two-stage fidelity-then-perception pipeline (Zhang et al., 2022), Bayesian optimization for dynamic loss weighting (Zhu et al., 2024), and multi-objective optimization strategies (Sun et al., 2024). GAN-based methods typically favor fidelity but produce over-smoothed outputs, whereas diffusion-based models generate sharp textures at the risk of hallucinations. Existing balancing techniques are mostly GAN-centric. Our approach bridges this gap by constraining T2I diffusion with spatially grounded semantic guidance, improving fidelity while preserving perceptual quality.

## 3 METHODOLOGY

### 3.1 MOTIVATION AND FRAMEWORK OVERVIEW

**Preliminary: Stable Diffusion.**    Our method builds on Stable Diffusion (SD), a latent diffusion model for T2I generation. SD operates in a compressed latent space for efficiency, where an autoencoder maps an image to latent $\mathbf{z}_0 = \mathcal{E}(I_0)$ and reconstructs it as $I_0 = \mathcal{D}(\mathbf{z}_0)$.

The forward diffusion gradually perturbs $\mathbf{z}_0$ with Gaussian noise:

$$q(\mathbf{z}_t \mid \mathbf{z}_{t-1}) = \mathcal{N}(\mathbf{z}_t; \sqrt{1 - \beta_t}\mathbf{z}_{t-1}, \beta_t\mathbf{I}), \quad t = 1, \ldots, T. \tag{1}$$

The reverse process recovers $\mathbf{z}_{t-1}$ from $\mathbf{z}_t$ via a denoising network $\epsilon_\theta$:

$$p_\theta(\mathbf{z}_{t-1} \mid \mathbf{z}_t) = \mathcal{N}(\mu_\theta(\mathbf{z}_t, t), \mathbf{\Sigma}_\theta(\mathbf{z}_t, t)), \tag{2}$$

which is trained to minimize the noise prediction objective:

$$\mathcal{L}_{SD} = \mathbb{E}_{\mathbf{z}_0, t, \epsilon}\left[\|\epsilon - \epsilon_\theta(\sqrt{\bar{\alpha}_t}\mathbf{z}_0 + \sqrt{1 - \bar{\alpha}_t}\epsilon, t)\|_2^2\right], \tag{3}$$

where $\epsilon \sim \mathcal{N}(0, \mathbf{I})$, $\bar{\alpha}_t = \prod_{i=0}^{t}(1 - \beta_i)$. Generation starts from noise $\mathbf{z}_T \sim \mathcal{N}(0, \mathbf{I})$ and iteratively applies reverse denoising to obtain $\mathbf{z}_0$.

**Motivation.**    Image SR faces the well-known perception-distortion trade-off (Blau & Michaeli, 2018): GAN-based methods (Zhang et al., 2021; Wang et al., 2021) achieve strong fidelity but often yield over-smoothed results lacking realistic details, while diffusion-based methods (Wang et al., 2024; Lin et al., 2024) generate sharper details and higher perceptual quality but frequently hallucinate content, leading to fidelity loss. This motivates our goal: harnessing the generative strength of diffusion while constraining it with spatial and semantic priors to reduce distortion.

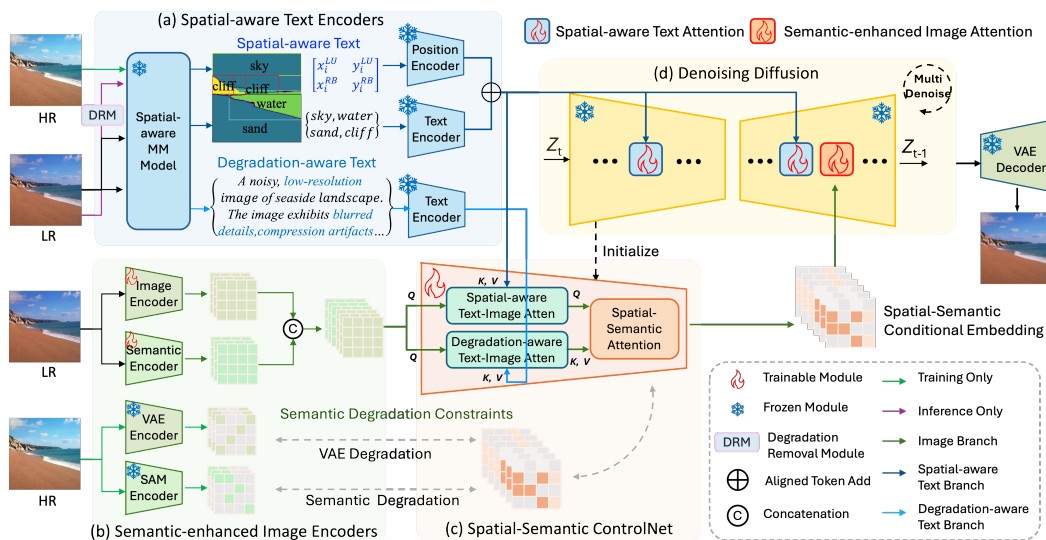

Figure 2: Framework overview. (a) Spatial-aware text encoders generate position-grounded textual prompts (Sec. 3.2); (b) Semantic-enhanced image encoders extract semantic-enhanced visual features with degradation constraints (Sec. 3.3); (c) Spatial-semantic ControlNet integrates these multimodal conditions (Sec. 3.4); (d) Spatial-semantic guided diffusion fuses semantic and spatial guidance with generative priors (Sec. 3.5).

**Framework Overview.** We propose SpaSemSR, a Spatial-Semantic guided SR framework that steers diffusion with two complementary forms of guidance (Fig. 2): **(a)** *Spatial-grounded textual guidance*, which integrates semantic prompts with object-level spatial cues, improving fidelity by aligning text semantics with visual structure (Sec. 3.2); **(b)** *Semantic-enhanced visual guidance*, which extracts semantic-enhanced features from degraded LR inputs under semantic degradation constraints, improving perceptual realism (Sec. 3.3). These guidances are fused in **(c)** the proposed *Spatial-Semantic ControlNet* (Sec. 3.4) via parallel cross-attention and integrated into **(d)** our *Spatial-Semantic guided Diffusion model* (Sec. 3.5), where they adaptively modulate the generative prior. By integrating spatial-grounded textual and semantic-enhanced visual guidance, SpaSemSR achieves reconstructions that are both perceptually realistic and faithful to the ground truth, effectively balancing the perception-distortion trade-off.

**Model Variants.** To validate the contributions of the proposed *spatial-grounded textual guidance* and *semantic-enhanced visual guidance*, we design four variants: (i) **Fidelity**: retains only spatial-grounded textual guidance with image-encoder constraints, without semantic-encoder constraints; (ii) **Perception**: employs semantic-enhanced visual guidance (image + semantic encoders and constraints), without spatial grounding; (iii) **Balance-1**: combines spatial-grounded textual guidance with semantic-encoder constraints; (iv) **Balance-2**: full model combining spatial-grounded textual guidance with both image and semantic encoder constraints. Detailed in the Appendix B.

### 3.2 SPATIAL-AWARE TEXT ENCODERS

In conventional T2I models, text inputs guide diffusion models but lack spatial grounding. As a result, textual prompts may misalign semantics with image regions, leading to reduced fidelity and unwanted distortions. To address this, we propose spatial-aware text encoders that integrate both spatially grounded semantics and degradation-aware information.

**Spatial-grounded Text Representations.** We introduce a spatial-aware text fusion mechanism that combines object semantics with their spatial locations, yielding more faithful region-to-text alignment. Specifically, given a low-resolution image $I_{LR}$, we employ a pretrained Grounded-SAM (Ren et al., 2024) to extract object-level textual tags $\mathbf{x}_{\text{obj-text}}$ and corresponding bounding boxes $\mathbf{x}_{\text{spa}}$:

$$\mathbf{x}_{\text{obj-text}} = \mathcal{F}_{\text{obj-text}}(I_{LR}), \quad \mathbf{x}_{\text{spa}} = \mathcal{F}_{\text{spa}}(I_{LR}), \tag{4}$$

where $\mathcal{F}_{\text{obj-text}}$ and $\mathcal{F}_{\text{spa}}$ are the object recognition and bounding box model from the pretrained Grounded-SAM.

The spatial coordinates are encoded with sinusoidal positional encoding $\mathcal{F}_{\text{PE}}(\cdot)$ to obtain embeddings $\mathbf{e}_{\text{spa}}$, which are then fused object-wise with textual embeddings $\mathbf{e}_{\text{obj-text}}$ from the pretrained CLIP encoder $\mathcal{E}_{\text{text}}$ (Radford et al., 2021):

$$\mathbf{e}_{\text{spa}} = \mathcal{F}_{\text{PE}}(\mathbf{x}_{\text{spa}}), \quad \mathbf{e}_{\text{obj-text}} = \mathcal{E}_{\text{text}}(\mathbf{x}_{\text{obj-text}}), \quad \mathbf{e}_{\text{spa-text}} = \mathcal{F}_{\text{fusion}}(\mathbf{e}_{\text{obj-text}}, \mathbf{e}_{\text{spa}}), \tag{5}$$

where $\mathcal{F}_{\text{fusion}}(\cdot)$ is the object-wise fusion function. This ensures a unique semantic-spatial correspondence, producing spatial-grounded text embeddings $\mathbf{e}_{\text{spa-text}}$ that better preserve fidelity.

**Degradation-aware Text Representations.** Object-level tags capture semantics but not degradation characteristics (e.g., blur, noise, compression). To complement them, we adopt degradation-aware textual priors (Qu et al., 2024) by using LLaVA (Liu et al., 2023) to generate an image-level description $\mathbf{x}_{\text{deg-text}}$ that encodes coarse attributes such as sharpness and noise:

$$\mathbf{x}_{\text{deg-text}} = \mathcal{F}_{\text{deg-text}}(I_{LR}), \quad \mathbf{e}_{\text{deg-text}} = \mathcal{E}_{\text{text}}(\mathbf{x}_{\text{deg-text}}), \tag{6}$$

where $\mathcal{F}_{\text{deg-text}}$ is the pretrained degradation-aware textual generation model. Thus, for each $I_{LR}$, we obtain two complementary textual priors: (i) spatial-grounded semantic prompts $\mathbf{e}_{\text{spa-text}}$ that enhance region-level semantics, and (ii) degradation-aware prompts $\mathbf{e}_{\text{deg-text}}$ that facilitate degradation modeling. Finally, these dual embeddings are jointly fed into ControlNet and the diffusion model, enabling high-fidelity, spatially consistent, and degradation-aware textual representations.

## 3.3 SEMANTIC-ENHANCED IMAGE ENCODERS

Image SR aims to reconstruct an HR image from a degraded LR input. Diffusion-based SR models typically rely on LR images as control conditions. However, severe degradation often destroys local structures, leading to ambiguous or misleading semantics.

To address this, we design a dual-encoder system with our semantic degradation loss that extracts complementary features from the LR input: (1) a low-level encoder $\mathcal{E}_{\text{img}}$ that captures latent structural details, and (2) a high-level semantic encoder $\mathcal{E}_{\text{sem}}$ that emphasizes semantic consistency. The perception ability of these features is further reinforced by semantic degradation constraints derived from pretrained VAE (Rombach et al., 2022) and SAM (Kirillov et al., 2023) encoders. This design produces semantic-enhanced features that are robust to degradation and preserve meaningful visual semantics. Formally, given an LR image $I_{LR}$, the two encoders extract high-level semantic features $\mathbf{x}_{\text{sem}}$ and low-level latent features $\mathbf{x}_{\text{img}}$, respectively:

$$\mathbf{x}_{\text{img}} = \mathcal{E}_{\text{img}}(I_{LR}), \quad \mathbf{x}_{\text{sem}} = \mathcal{E}_{\text{sem}}(I_{LR}), \quad \mathbf{x}_{\text{sem-img}} = \text{Concat}(\mathbf{x}_{\text{img}}, \mathbf{x}_{\text{sem-img}}), \tag{7}$$

where $\text{Concat}(\cdot)$ denotes the concatenation operation. The fused feature $\mathbf{x}_{\text{sem-img}}$ is injected into each ControlNet layer. Following (Qu et al., 2024), we extract a hybrid representation $\mathbf{x}_{\text{hyb}}^i$ from the $i$-th layer and evenly split it into two streams:

$$\mathbf{x}_{\text{hyb}}^i = \text{ControlNet}^i(\mathbf{x}_{\text{sem-img}}), \quad [\mathbf{x}_{\text{img}}^i, \mathbf{x}_{\text{sem}}^i] = \text{Split}(\mathbf{x}_{\text{hyb}}^i), \tag{8}$$

where $\mathbf{x}_{\text{img}}^i$ and $\mathbf{x}_{\text{sem}}^i$ retain the respective low-level and semantic branches, ensuring disentangled feature learning.

**Semantic Degradation Loss.** To guide each encoder, we introduce a semantic degradation loss:

$$\mathcal{L}_{\text{ControlNet}}^{\text{SemDeg}} = \sum_{i=1}^{n} [(1-\lambda)\|\mathbf{x}_{\text{img}}^i - \hat{\mathbf{x}}_{\text{img}}^i\|_1 + \lambda\|\mathbf{x}_{\text{sem}}^i - \hat{\mathbf{x}}_{\text{sem}}^i\|_1], \tag{9}$$

where $\hat{\mathbf{x}}_{\text{img}}^i$ and $\hat{\mathbf{x}}_{\text{sem}}^i$ are the reference features extracted from pretrained VAE and SAM encoders on the HR image, respectively. The balancing coefficient $\lambda \in [0, 1]$ controls the relative emphasis on structural fidelity versus semantic consistency. By enforcing this loss, each encoder is encouraged to specialize: low-level encoders align with structural details, while semantic encoders capture high-level semantics, thereby providing more reliable features to guide the diffusion model under complex degradations.

## 3.4 SPATIAL-SEMANTIC CONTROLNET

To fully leverage semantic-enhanced image priors and spatial-grounded textual knowledge, we design a Spatial-Semantic ControlNet, which serves as a controller integrated with Stable Diffusion.

Specifically, we introduce two parallel and one fusion attention modules: *Spatial-aware Text-Image Attention* (SpaAtten) and *Degradation-aware Text-Image Attention* (DegAtten), followed by a *Spatial-Semantic Attention* (SpaSemAtten) fusion layer. As shown in Fig. 2, the spatial-aware text features $\mathbf{e}_{\text{spa-text}}$ and degradation-aware text features $\mathbf{e}_{\text{deg-text}}$, obtained from Sec. 3.2, serve as the key-value sets for SpaAtten and DegAtten, respectively. The semantic-enhanced image features $\mathbf{x}_{\text{sem-img}}$ (Sec. 3.3) are shared as queries across both branches. The outputs of SpaAtten and DegAtten are then fused by SpaSemAtten to integrate complementary cues. Formally, the attention operations are defined as:

$$\mathbf{y}_{\text{spa}} = \text{SpaAtten}(\mathbf{x}_{\text{sem-img}}\mathbf{W}_{\text{spa}}^Q, \mathbf{e}_{\text{spa-text}}\mathbf{W}_{\text{spa}}^K, \mathbf{e}_{\text{spa-text}}\mathbf{W}_{\text{spa}}^V),$$

$$\mathbf{y}_{\text{deg}} = \text{DegAtten}(\mathbf{x}_{\text{sem-img}}\mathbf{W}_{\text{deg}}^Q, \mathbf{e}_{\text{deg-text}}\mathbf{W}_{\text{deg}}^K, \mathbf{e}_{\text{deg-text}}\mathbf{W}_{\text{deg}}^V), \tag{10}$$

$$\mathbf{x}_{\text{spa-sem}} = \text{SpaSemAtten}(\mathbf{y}_{\text{spa}}\mathbf{W}_{\text{SpaSem}}^Q, \mathbf{y}_{\text{deg}}\mathbf{W}_{\text{SpaSem}}^K, \mathbf{y}_{\text{deg}}\mathbf{W}_{\text{SpaSem}}^V),$$

where $\mathbf{y}_{\text{spa}}$ and $\mathbf{y}_{\text{deg}}$ are intermediate outputs from the two branches, and $\mathbf{x}_{\text{spa-sem}}$ denotes the fused representation. Each cross-attention follows the standard formulation: $\text{Atten}(Q, K, V) = \text{Softmax}\left(\frac{QK^T}{\sqrt{d}}\right)V$. This design allows SpaAtten to inject precise spatial textual cues while DegAtten introduces degradation-aware textual context. The subsequent SpaSemAtten layer fuses both signals, balancing local spatial fidelity with global semantic consistency. Therefore, the diffusion model generates contents that are not only perceptually realistic but also structurally faithful.

### 3.5 DIFFUSION VIA SPATIAL-SEMANTIC GUIDANCE

To seamlessly integrate cross-modal semantic and spatial priors with T2I generative priors, we introduce the spatial-semantic guided diffusion model (*SpaSemDM*). SpaSemDM adaptively learns to integrate semantic and spatial cues into the denoising process, thereby producing reconstructions that are both perceptually compelling and faithful to the source content.

In SpaSemDM, the denoising network is augmented with two control conditions: (1) spatially aware textual features $\mathbf{e}_{\text{spa-text}}$ extracted from Spatial-Aware Text Encoders, and (2) spatial-semantic visual embeddings $\mathbf{x}_{\text{spa-sem}}$ from the Spatial-Semantic ControlNet. To fuse these conditions, we design two additional attention modules: *Spatial-aware Text Attention* (SpaTextAtten) and *Semantic-enhanced Image Attention* (SemImgAtten), which jointly inject spatial grounding and semantic alignment into the latent space during diffusion learning. This mechanism allows SpaSemDM to better navigate the perception-distortion trade-off.

**Training and Optimization.** During training, we first obtain the latent representation $z_0$ of an HR image, which is progressively corrupted by Gaussian noise to yield $z_t$ at step $t$. Conditioned on $t$, the LR input $I_{\text{LR}}$, its degradation-aware text prompt $\mathbf{x}_{\text{deg-text}}$, and the spatial-grounded text prompt $\{\mathbf{x}_{\text{obj-text}}, \mathbf{x}_{\text{spa}}\}$, SpaSemDM network $\epsilon_\theta$ is trained to predict the noise added to $z_t$. The objective is:

$$\mathcal{L}_{SD}^{\text{SpaSem}} = \mathbb{E}_{z_0, t, I_{\text{LR}}, \epsilon}\left[\|\epsilon - \epsilon_\theta(\mathbf{z}_t, t; I_{\text{LR}}, \mathbf{x}_{\text{deg-text}}, \{\mathbf{x}_{\text{obj-text}}, \mathbf{x}_{\text{spa}}\})\|_2^2\right]. \tag{11}$$

In our model training, the final loss can be expressed as:

$$\mathcal{L} = \mathcal{L}_{SD}^{\text{SpaSem}} + \lambda_{\text{ControlNet}}\mathcal{L}_{\text{ControlNet}}^{\text{SemDeg}}, \tag{12}$$

where $\mathcal{L}_{\text{ControlNet}}^{\text{SemDeg}}$ regularizes the image encoders and ControlNet to enhance semantic fidelity, and $\lambda_{\text{ControlNet}}$ is the weighting coefficient.

## 4 EXPERIMENTS

**Implementation.** Our framework is built on ControlNet (Zhang et al., 2023) with Stable Diffusion v2 (Rombach et al., 2022) as the backbone. Semantic features from a pretrained SAM image encoder and image features from a VAE encoder serve as constraints for the corresponding encoders. During training, Grounded-SAM (Ren et al., 2024) provides positional information and object-level tags from HR images. For training inference consistency, LR inputs in inference are processed by a degradation removal model (DRM) to restore a clean image following (Chen et al., 2025) , and then bounding boxes are extracted from this restored image with Grounded-SAM. Training runs for 200k iterations (batch size 32, lr $5 \times 10^{-5}$) at $512 \times 512$ resolution on 4×RTX 6000 GPUs for three days.

Table 1: Comparison with GAN methods.

| Dataset | Model | Reference Fidelity | | Non-reference Perception | | | [a] |
|---|---|---|---|---|---|---|---|
| | | PSNR↑ | SSIM↑ | CLIP-IQA↑ | MUSIQ↑ | MANIQA↑ | |
| DIV2K-Val | BSRGAN | 21.74 | 0.5530 | 0.5234 | 59.16 | 0.3528 | |
| | Real-ESRGAN | 21.86 | 0.5746 | 0.5485 | 58.80 | 0.3776 | |
| | SwinIR | 21.45 | 0.5639 | 0.5467 | 59.03 | 0.3634 | |
| | **Ours-Fidelity** | 21.48 | 0.5445 | 0.6523 | 61.88 | 0.4520 | |
| | **Ours-Perception** | 20.70 | 0.5204 | 0.7194 | 69.32 | 0.5575 | |
| | **Ours-Balance1** | 21.51 | 0.5423 | 0.6609 | 62.14 | 0.4612 | |
| | **Ours-Balance2** | 21.31 | 0.5340 | 0.6932 | 63.32 | 0.4945 | |
| RealSR | BSRGAN | 26.38 | 0.7651 | 0.5112 | 63.28 | 0.3754 | |
| | Real-ESRGAN | 25.69 | 0.7614 | 0.4490 | 60.37 | 0.3730 | |
| | SwinIR | 26.31 | 0.7729 | 0.4364 | 58.69 | 0.3444 | |
| | **Ours-Fidelity** | 26.02 | 0.7434 | 0.5678 | 61.70 | 0.4276 | |
| | **Ours-Perception** | 23.90 | 0.6897 | 0.6900 | 69.79 | 0.5759 | |
| | **Ours-Balance1** | 25.84 | 0.7312 | 0.6116 | 64.03 | 0.4736 | |
| | **Ours-Balance2** | 25.74 | 0.7306 | 0.5893 | 62.96 | 0.4637 | |
| DrealSR | BSRGAN | 28.70 | 0.8028 | 0.5093 | 57.16 | 0.3447 | |
| | Real-ESRGAN | 28.61 | 0.8052 | 0.4517 | 54.27 | 0.3448 | |
| | SwinIR | 28.50 | 0.8044 | 0.4445 | 52.74 | 0.3298 | |
| | **Ours-Fidelity** | 29.14 | 0.7968 | 0.5719 | 54.64 | 0.3825 | |
| | **Ours-Perception** | 26.72 | 0.7406 | 0.6930 | 65.37 | 0.5308 | |
| | **Ours-Balance1** | 29.21 | 0.7945 | 0.5728 | 55.60 | 0.4023 | |
| | **Ours-Balance2** | 28.97 | 0.7826 | 0.5630 | 55.79 | 0.3998 | |

[a]Highlight **Best**, Second-best, *Third-best*, Top-4 .

Table 2: Comparison with diffusion methods.

| Dataset | Model | Reference Fidelity | | Non-reference Perception | | |
|---|---|---|---|---|---|---|
| | | PSNR↑ | SSIM↑ | CLIP-IQA↑ | MUSIQ↑ | MANIQA↑ |
| DIV2K-Val | StableSR | 20.74 | 0.4888 | 0.6605 | 63.19 | 0.4002 |
| | DiffBIR | 20.57 | 0.4740 | 0.7359 | 69.93 | 0.5763 |
| | PASD | 20.77 | 0.5022 | 0.6140 | 63.29 | 0.4581 |
| | SeeSR | 21.00 | 0.5362 | 0.7074 | 68.81 | 0.5149 |
| | XPSR | 20.56 | 0.5081 | 0.7826 | 70.07 | 0.6108 |
| | **Ours-Fidelity** | 21.48 | 0.5445 | 0.6523 | 61.88 | 0.4520 |
| | **Ours-Perception** | 20.70 | 0.5204 | 0.7194 | 69.32 | 0.5575 |
| | **Ours-Balance1** | 21.51 | 0.5423 | 0.6609 | 62.14 | 0.4612 |
| | **Ours-Balance2** | 21.31 | 0.5340 | 0.6932 | 63.32 | 0.4945 |
| RealSR | StableSR | 24.70 | 0.7085 | 0.6166 | 65.18 | 0.4178 |
| | DiffBIR | 24.83 | 0.6501 | 0.7054 | 69.28 | 0.5596 |
| | PASD | 25.26 | 0.7191 | 0.6249 | 67.78 | 0.4971 |
| | SeeSR | 25.15 | 0.7210 | 0.6704 | 69.82 | 0.5395 |
| | XPSR | 23.74 | 0.6734 | 0.7417 | 71.45 | 0.6293 |
| | **Ours-Fidelity** | 26.02 | 0.7434 | 0.5678 | 61.70 | 0.4276 |
| | **Ours-Perception** | 23.90 | 0.6897 | 0.6900 | 69.79 | 0.5759 |
| | **Ours-Balance1** | 25.84 | 0.7312 | 0.6116 | 64.03 | 0.4736 |
| | **Ours-Balance2** | 25.74 | 0.7306 | 0.5893 | 62.96 | 0.4637 |
| DrealSR | StableSR | 28.07 | 0.7489 | 0.6375 | 58.99 | 0.3892 |
| | DiffBIR | 25.90 | 0.6245 | 0.7068 | 66.13 | 0.5526 |
| | PASD | 27.07 | 0.7251 | 0.6710 | 64.56 | 0.5061 |
| | SeeSR | 28.07 | 0.7684 | 0.6911 | 65.09 | 0.5115 |
| | XPSR | 26.55 | 0.7289 | 0.7433 | 67.02 | 0.5684 |
| | **Ours-Fidelity** | 29.14 | 0.7968 | 0.5719 | 54.64 | 0.3825 |
| | **Ours-Perception** | 26.72 | 0.7406 | 0.6930 | 65.37 | 0.5308 |
| | **Ours-Balance1** | 29.21 | 0.7945 | 0.5728 | 55.60 | 0.4023 |
| | **Ours-Balance2** | 28.97 | 0.7826 | 0.5630 | 55.79 | 0.3998 |

**Datasets and Metrics.** *Training Data:* We train on DIV2K (Agustsson & Timofte, 2017), DIV8K (Gu et al., 2019b), Flickr2K (Timofte et al., 2017), OutdoorSceneTraining (Kim & Son, 2021), Unsplash2K (Wang et al., 2018), and 5k FFHQ faces (Karras et al., 2019), using Real-ESRGAN's (Wang et al., 2021) degradation pipeline to synthesize LR-HR pairs. *Test Data:* Synthetic evaluation uses 3,000 degraded DIV2K validation patches with the same pipeline as Real-ESRGAN. For real-world data, we follow StableSR (Wang et al., 2024), evaluate on RealSR (Cai et al., 2019) and DRealSR (Wei et al., 2020), center-cropped to 128×128 LR images. We also evaluate on RealLR200 (Wu et al., 2024c), which lacks ground-truth. *Metrics:* For distortion, we report PSNR and SSIM (Wang et al., 2004). For perceptual quality, we adopt CLIP-IQA (Wang et al., 2023), MUSIQ (Ke et al., 2021), and MANIQA (Yang et al., 2022).

## 4.1 BALANCING PERCEPTION AND DISTORTION

A key challenge in SR is reconciling the perception-distortion trade-off. We evaluate SpaSemSR with its fidelity-, perception-, and balance-oriented variants against representative GAN-based methods (Real-ESRGAN (Wang et al., 2021), BSRGAN (Zhang et al., 2021), SwinIR (Liang et al., 2021)) and diffusion-based methods (StableSR (Wang et al., 2024), DiffBIR (Lin et al., 2024), PASD (Yang et al., 2024), SeeSR (Wu et al., 2024c), XPSR (Qu et al., 2024)).

**Quantitative Comparisons.** **(1) Perceptual Quality:** As shown in Table 1, in terms of perception, our four variants significantly outperform GAN-based methods on CLIP-IQA and MANIQA, and overall deliver higher MUSIQ scores than the GAN-based methods. When compared against diffusion-based methods, our perception variant secures the third-best results in Table 2. These results indicate that ours outperform GAN-based methods and are comparable to diffusion-based methods in perceptual quality. **(2) Reconstruction Fidelity:** Table 2 indicates that our *fidelity*, *balance-1*, and *balance-2* variants outperform the diffusion-based methods from the perspective of fidelity. These three variants are within a minor gap of GAN-based methods and even surpass them on the DRealSR real-world images in terms of PSNR in Table 1. These results demonstrate that our model can effectively enhance the fidelity of diffusion-based models and achieve high fidelity comparable to GAN-based methods. **(3) Trade-off analysis:** Across both tables, GAN-based methods generally outperform diffusion-based methods on reference distortion metrics like PSNR and SSIM, but lag far behind diffusion-based methods on non-reference perceptual metrics. Ours balance the quantitative behavior of GAN-based methods and diffusion-based methods while mitigating their respective weaknesses, achieving a stronger perception-distortion trade-off.

**Qualitative Comparisons.** Visual comparisons (Fig. 3) further validate our approach. **(1) Perceptual Quality:** GAN-based methods typically produce over-smoothed or blurry reconstructions that

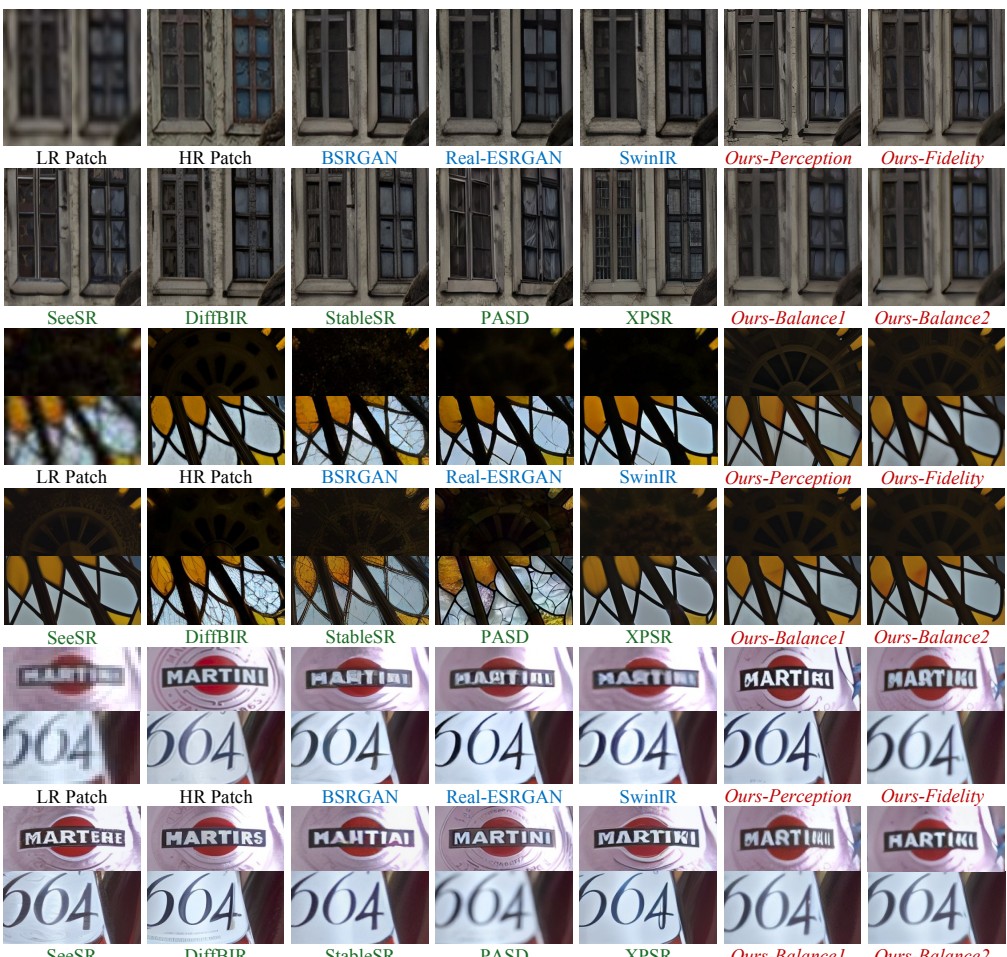

Figure 3: Qualitative comparisons with different methods. Zoom in for a better view.

lack human-perceived realism. In the case of windows from Fig. 3, they struggle to generate plausible window frame textures, resulting in overly-smoothed or blurry reconstructions that fall short in texture richness and sharpness compared to diffusion models, resulting in inferior perceptual quality. In contrast, our four variants reconstruct clearer window-frame textures and sharper, perceptually plausible details, yielding higher perceived quality than GAN-based methods. **(2) Reconstruction Fidelity:** Although diffusion-based methods can generate clear textures and details, their inherent generative diversity often results in details and textures inconsistent with the ground truth. For example, SeeSR and DiffBIR produce inconsistent window frame textures, while StableSR, PASD, and XPSR generate extra curtains or window frames on originally reflective window glass. This diversity-driven hallucination undermines fidelity, causes semantic misalignment with the ground truth. In comparison, our four variants produce images with realistic, fine-grained textures that closely match the ground truth, exhibiting minimal distortion and substantially fewer hallucinated artifacts. **(3) Trade-off analysis:** Our methods deliver clearer, fine-grained textures than GAN-based methods, achieve higher perceptual quality, and exhibit fewer hallucinations with better ground truth alignment than diffusion-based methods, producing high-fidelity reconstructions with a well-balanced perception-distortion trade-off. More qualitative results are in Appendix C.3.

## 4.2 ABLATION STUDY

We conduct ablation experiments to evaluate the contributions of the proposed *spatial-grounded textual guidance* and *semantic-enhanced visual guidance*. Fig. 4 displays the content of spatial-grounded textual and semantic-enhanced visual guidance. Table 3 summarizes results for four model variants across synthetic and real-world datasets. These variants enable systematic isolation and quantification of each guidance component's effect on fidelity, perception, and overall balance.

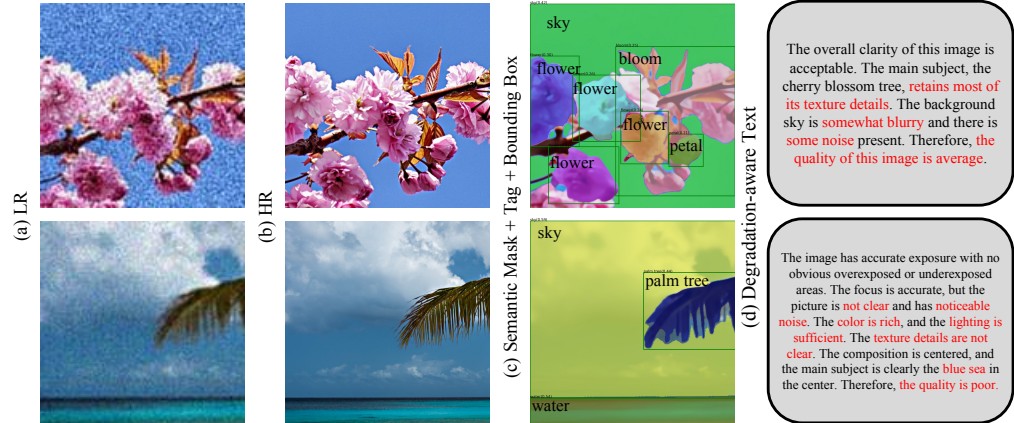

Figure 4: Visualization of spatial-grounded textual and semantic-enhanced visual guidance.

Table 3: Ablation study, highlights the Best performance.

| | Vision Encoder | | Degradation Constraint | | Multimodal Encoder | | RealSR | | | | | DrealSR | | | | | DIV2K-Val | | | | |
|---|---|---|---|---|---|---|---|---|---|---|---|---|---|---|---|---|---|---|---|---|---|
| | Image | Semantic | VAE | Semantic | Spatial | Text | PSNR↑ | SSIM↑ | CLIP-IQA↑ | MUSIQ↑ | MANIQA↑ | PSNR↑ | SSIM↑ | CLIP-IQA↑ | MUSIQ↑ | MANIQA↑ | PSNR↑ | SSIM↑ | CLIP-IQA↑ | MUSIQ↑ | MANIQA↑ |
| Ours-Fidelity | ✓ | | ✓ | | ✓ | ✓ | **26.02** | **0.7434** | 0.5678 | 61.70 | 0.4276 | 29.14 | **0.7968** | 0.5719 | 54.64 | 0.3825 | 21.48 | **0.5445** | 0.6523 | 61.88 | 0.4520 |
| Ours-Balance1 | | ✓ | | ✓ | ✓ | ✓ | 25.84 | 0.7312 | **0.6116** | **64.03** | **0.4736** | **29.21** | 0.7945 | **0.5728** | **55.60** | **0.4023** | **21.51** | 0.5423 | **0.6609** | **62.14** | **0.4612** |
| Ours-Perception | ✓ | ✓ | ✓ | ✓ | | ✓ | 23.90 | 0.6897 | **0.6900** | **69.79** | **0.5759** | 26.72 | 0.7406 | **0.6930** | **65.37** | **0.5308** | 20.70 | 0.5204 | **0.7194** | **69.32** | **0.5575** |
| Ours-Balance2 | ✓ | ✓ | ✓ | ✓ | ✓ | ✓ | **25.74** | **0.7306** | 0.5893 | 62.96 | 0.4637 | **28.97** | **0.7826** | 0.5630 | 55.79 | 0.3998 | **21.31** | **0.5340** | 0.6932 | 63.32 | 0.4945 |

**Guidance Visualization** In Fig. 4(c) and (d), the object-level tags with their corresponding bounding boxes, together with the degradation-aware text, constitute the spatial-grounded textual guidance. And the semantic mask in Fig. 4(c) serves as prior information of our semantic-enhanced visual guidance. Incorporating high-level semantic priors extracted from the SAM image encoder as guidance contributes to improving perceptual quality. In addition, guiding the model to align the visual semantic mask with the corresponding object-level tags further enhances image fidelity, leading to a better trade-off between perceptual quality and distortion. More results are in Appendix C.2.

**Effectiveness of Spatial-grounded textual guidance.** We compare *Balance-2* with *Perception*, which differs only by excluding spatial information. Table 3 confirms that adding spatial grounding improves fidelity metrics (PSNR/SSIM), though with a slight trade-off in perceptual quality scores. This trade-off is acceptable given SR's primary objective of preserving structural consistency. Moreover, the *Fidelity* and *Balance-1* variants, which both incorporate spatial information, consistently reduce hallucinations and outperform the *Perception* variant on fidelity measures, validating that spatial information effectively aligns semantic prompts with visual structure.

**Effectiveness of Semantic-enhanced Visual Guidance.** We next assess the impact of semantic-enhanced visual guidance by comparing the *Fidelity*, *Balance-1*, and *Balance-2* variants with different degradation constraints. As shown in Table 3, variants equipped with semantic constraints achieve clear improvements on perceptual metrics such as MUSIQ, MANIQA, and CLIP-IQA, indicating that high-level semantic priors enrich perceptual realism and are robust to LR degradation. In contrast, the *Fidelity* variant, constrained only by the VAE encoder, achieves slightly higher PSNR/SSIM since VAE features emphasize pixel-level accuracy. *Balance-1* and *Balance-2* variants combining both spatial and semantic guidance yield the best overall perception-distortion balance, demonstrating complementarity of spatial-level fidelity and semantic-level realism. More results in Appendix C.1.

## 5 CONCLUSIONS

We present SpaSemSR, a spatial-semantic guided framework for image super-resolution that explicitly addresses the perception-distortion trade-off. By integrating spatial-grounded textual guidance with semantic-enhanced visual guidance, our approach leverages powerful diffusion priors while mitigating distortion and hallucination. The proposed spatial-semantic attention mechanisms enable adaptive fusion of multimodal priors, producing reconstructions that are both perceptually realistic and structurally faithful. Extensive experiments demonstrate that SpaSemSR attains state-of-the-art perception–distortion balance across multiple benchmarks by leveraging spatial and semantic guidance.

REPRODUCIBILITY STATEMENT

We have taken multiple steps to ensure the reproducibility of our results. A comprehensive description of our method in Sec. 3, and experimental setup is provided in the main paper in Sec. 4, with additional implementation details included in the Appendix A. Furthermore, we will release our implementation code and the trained model checkpoints upon acceptance.

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

# APPENDIX

## THE USE OF LARGE LANGUAGE MODELS

In the preparation of this paper, LLMs were used solely for polishing author-written text, including spell and grammar checks for editing purposes.

## A IMPLEMENTATION DETAILS

### A.1 SPATIAL-AWARE TEXT ENCODERS

**Spatial-aware Text Generation** To generate spatial-aware text, we employ Grounded-SAM (Ren et al., 2024) to obtain both tag and positional information. The positional information is embedded through a fixed position encoder, while the tag is embedded by a pre-trained CLIP text encoder (Radford et al., 2021), producing a unified representation for subsequent processing.

Grounded-SAM is a unified multimodal framework that performs automated object detection, segmentation, and generation from a single image input. It integrates Grounding DINO, Segment Anything, and Recognize Anything to achieve fully automated object detection and localization, and further incorporates Stable Diffusion to enable controllable image synthesis and editing within a unified pipeline. In our framework, we use Grounded-SAM to generate tags $\mathbf{x}_{\text{obj-text}}$ and bounding boxes $\mathbf{x}_{\text{spa}}$, represented as four-tuples corresponding to the top-left and bottom-right coordinates:

$$\mathbf{x}_{\text{spa}} = \begin{bmatrix} x_i^{LU} & y_i^{LU} \\ x_i^{RB} & y_i^{RB} \end{bmatrix}. \tag{13}$$

where $x_i^{\text{LU}}$ and $y_i^{\text{LU}}$ denote the horizontal and vertical coordinates of the left-upper (LU) corner of the $i$-th bounding box, and $x_i^{\text{RB}}$ and $y_i^{\text{RB}}$ denote the horizontal and vertical coordinates of its right-bottom (RB) corner. Next, we compute the spatial embedding of $\mathbf{x}_{\text{spa}}$ using the following sinusoidal positional encoding function:

$$PE(u, 2k) = \sin\left(\frac{u}{10000^{2k/(d/4)}}\right), \quad PE(u, 2k+1) = \cos\left(\frac{u}{10000^{2k/(d/4)}}\right). \tag{14}$$

where $u \in \{x_i^{\text{LU}}, y_i^{\text{LU}}, x_i^{\text{RB}}, y_i^{\text{RB}}\}$ denotes one of the four coordinates of the $i$-th bounding box, $k = 0, 1, \ldots, \frac{d}{8} - 1$ is the frequency index, and $d$ is the embedding dimension. We set the embedding dimension $d$ to $1024$, which matches the CLIP text embedding dimension. And the first $512$ dimensions are allocated to the left-upper coordinates and the remaining $512$ dimensions to the right-bottom coordinates, the final $\mathbf{e}_{\text{spa}}$ can be expessed as:

$$\mathbf{e}_{spa} = \begin{bmatrix} PE(x_i^{\text{LU}}) & PE(y_i^{\text{LU}}) & PE(x_i^{\text{RB}}) & PE(y_i^{\text{RB}}) \end{bmatrix} \in \mathbb{R}^{1024}. \tag{15}$$

Semantic tags $\mathbf{x}_{\text{obj-text}}$ will be go through a pre-trained CLIP text encoder to get textual embeddings $\mathbf{e}_{\text{obj-text}}$. Then an object-wise fusion function $\mathcal{F}_{\text{fusion}}$ is used to fuse two semantic tags embedding with spatial embedding to ensure a unique semantic-spatial correspondence:

$$\mathcal{F}_{\text{fusion}}(\{\mathbf{e}_{\text{obj-text}}^i, \mathbf{e}_{\text{spa}}^i\}_{i=1}^N) = \{\mathbf{e}_{\text{obj-text}}^i + \mathbf{e}_{\text{spa}}^i\}_{i=1}^N, \quad \mathbf{e}_{\text{obj-text}}^i, \mathbf{e}_{\text{spa}}^i \in \mathbb{R}^{1024}, \tag{16}$$

where $\mathbf{e}_{\text{obj-text}}^i$ denotes the textual embedding of the $i$-th object-level tag, and $\mathbf{e}_{\text{spa}}^i$ denotes the corresponding spatial embedding of the same object.

To ensure consistency between training and inference while leveraging large multimodal models, we adopt different strategies for constructing spatial-aware text prompts. During training, we apply Grounded-SAM to extract spatial information from HR images and get object-level tags from the LR input. During inference, following FaithDiff (Chen et al., 2025), a degradation removal model (DRM) first restores a clean version of the LR image. Grounded-SAM is then applied to this restored image to obtain spatial information, then combined with object-level tags to form the spatial-aware text prompt.

**Degration-aware Text Generation**   To complement the object-level semantic tags, which cannot capture image degradation characteristics, we follow the approach of (Qu et al., 2024) and leverage a Q-Instruct fine-tuned LLaVA (Wu et al., 2024a) model to generate degradation-aware text $\mathbf{x}_{\text{deg-text}}$ by prompting it with the instruction "Describe and evaluate the quality of the image." The resulting text is subsequently encoded using a pre-trained CLIP text encoder to obtain degradation-aware embeddings $\mathbf{e}_{\text{deg-text}}$, which are used together with the spatial-aware text embeddings $\mathbf{e}_{\text{spa-text}}$ to control the diffusion process.

## A.2   Semantic-enhanced Image Encoders

**Image Encoder and Semantic Encoder**   We use two trainable image encoders $\mathcal{E}$, which have the same structure as the image encoder in ControlNet (Zhang et al., 2023), to extract the semantic-enhanced visual feature. For our task, we first upscale each LR image from $128 \times 128$ to $512 \times 512$, ensuring the resolution aligns with the input size of Stable Diffusion. The up-scaled image is then mapped into a $64 \times 64$ feature-space representation that matches the latent resolution of Stable Diffusion. Specifically, the trainable image encoder is composed of a sequence of $3 \times 3$ convolutional layers with SiLU activations and progressively increasing channel dimensions $(16, 32, 96, 256)$. Each stage of the encoder contains one stride-1 convolution for feature refinement, followed by one stride-2 convolution for spatial downsampling, thereby reducing the resolution from $512 \rightarrow 256 \rightarrow 128 \rightarrow 64$.

To adapt to the subsequent ControlNet architecture, we first concatenate the features obtained from the two image encoders, and then apply a single convolution layer to reduce the channel by half. Then we split the feature maps, which are extracted from the $i$-th layer of ControlNet, into two groups, corresponding to the low-channel image features and the high-channel semantic features. Each group is further processed by a single convolution layer to map them into the desired feature shape, yielding $\mathbf{x}_{\text{img}}^i \in \mathbb{R}^{H_i \times W_i \times 4}$, $\mathbf{x}_{\text{sem}}^i \in \mathbb{R}^{H_i \times W_i \times 256}$. Finally, we obtain the corresponding VAE features $\hat{\mathbf{x}}_{\text{img}}^i$ and semantic features $\hat{\mathbf{x}}_{\text{sem}}^i$ from the HR image, and impose an $\ell_1$ loss to enforce consistency to encourage specialization of each image encoder, providing low-level structural details and high-level semantics. The semantic degradation loss can be expressed as:

$$\mathcal{L}_{\text{ControlNet}}^{\text{SemDeg}} = \sum_{i=1}^{n=3} [(1-\lambda)\|\mathbf{x}_{\text{img}}^i - \hat{\mathbf{x}}_{\text{img}}^i\|_1 + \lambda\|\mathbf{x}_{\text{sem}}^i - \hat{\mathbf{x}}_{\text{sem}}^i\|_1], \tag{17}$$

## B   Variants Configuration

In the section 3.1 of the main paper, we proposed four variants: fidelity, Perceptual, Balance 1, and Balance 2. Each variant is designed to emphasize either fidelity or perceptual quality, or to achieve a better trade-off between the two. The configurations of these four variants are summarized as follows:

**(i) Fidelity:**   In this fidelity variant, we keep the whole spatial-aware text encoders; regarding semantic-enhanced image encoders, we only use the VAE encoder to constrain image encoder. Without incorporating semantic degradation constraints and a semantic encoder, the degradation loss is defined as follows:

$$\mathcal{L}_{\text{ControlNet}}^{\text{Deg}} = \sum_{i=1}^{n=3} \lambda\|\mathbf{x}_{\text{img}}^i - \hat{\mathbf{x}}_{\text{img}}^i\|_1, \tag{18}$$

**(ii) Perception:**   In this perception variant, we modify the spatial-aware text encoders by removing the spatial information $\mathbf{x}_{\text{spa}}$, and only keep object-level semantic tags $\mathbf{x}_{\text{obj-text}}$ and degradation-aware text $\mathbf{x}_{\text{deg-text}}$. Both the SAM image encoder and the VAE image encoder are employed to constrain the semantic encoder and the image encoder simultaneously, encouraging the model to learn high-level semantic representations, thereby enhancing perceptual quality.

**(iii) Balance-1:**   In this balanced variant, we retain the full spatial-aware text encoders, while removing the VAE-based degradation constraint and the image encoder. Instead, only the semantic degradation constraints and a semantic image encoder are applied, encouraging the model to learn

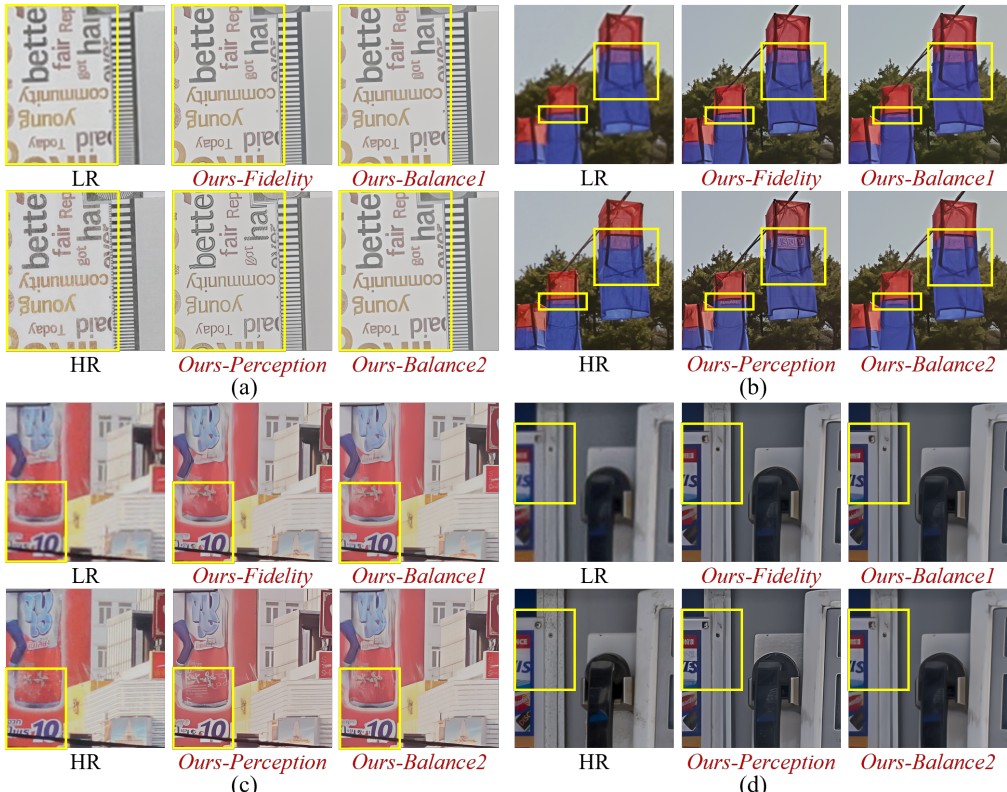

Figure 5: Ablation visualization. Zoom in for a better view.

high-level semantic representations and thereby yielding a balanced variant biased toward perceptual quality.

**(iv) Balance-2:** In the second balanced variant, we retain all modules, including the spatial-aware text encoders and the semantic-enhanced image encoders. This variant fully exploits both spatial-grounded textual guidance and semantic-enhanced visual guidance, aiming to achieve more favorable balance between perception and distortion.

## C  ADDITIONAL EXPERIMENTAL RESULTS

In this section, we present additional ablation visualization results, some visualization cases of spatial-grounded textual and semantic-enhanced visual guidance, and more qualitative results.

### C.1  ADDITIONAL ABLATION STUDY RESULTS

Fig. 5 shows the visualization results with different variants on real-world datasets. The *Perception* variant often introduces hallucinated textures (highlighted in yellow boxes), such as dashed strokes(a), artificial characters appearing on the lantern surface(b), overly detailed and fabricated surface patterns(c), and distorted letter shapes(d), which are inconsistent with the ground truth. In contrast, our *Fidelity*, *Balance-1*, and *Balance-2* variants significantly suppress these artifacts and better align with the ground truth through the spatial-grounded textual guidance.

### C.2  VISUALIZATION OF SPATIAL-GROUNDED TEXTUAL AND SEMANTIC-ENHANCED VISUAL GUIDANCE

Fig. 6 displays the spatial-grounded textual and semantic-enhanced visual guidance, including images processed by the DRM module, the object mask and its object-level tag, and corresponding bounding

boxes generated by Grounded-SAM, as well as the degradation-aware text generated by LLaVA, under different images. In addition, it also illustrates the tags directly extracted from the LR image using the degradation-aware prompt extractor (DAPE) proposed by SeeSR (Wu et al., 2024c). Considering that the object-level tags extracted by Grounded-SAM from the LR image, or even from the DRM-processed image, are not always satisfactory, another optional strategy in our experiments is to employ the DAPE module to extract object-level tags directly from the LR image, and then pair them with the corresponding bounding boxes. These components together constitute a spatial-aware text guidance that aligns spatial information with semantic textual prompts to enhance image fidelity.

## C.3   MORE QUALITATIVE COMPARISONS

Fig. 7 and Fig. 8 present additional visual comparison of all evaluated models on DIV2K-Val datasets. Fig. 9, Fig. 10, Fig. 11, Fig. 12 shows visual comparison on RealLR200 dataset, which do not provide the ground truth.

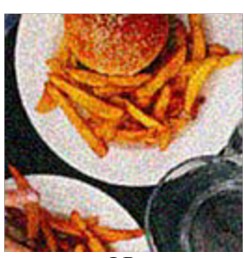 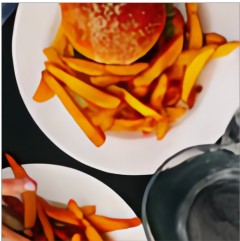 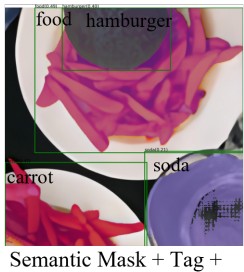

LR            DRM-processed Image            Semantic Mask + Tag +            Tags(DAPE)
                                              Bounding Box

cheeseburger,
table, plate, food,
French fries,
hamburger,
platter, soda,

The image has accurate exposure with no obvious overexposed or underexposed areas. The focus is accurate, but the picture is not clear and has noticeable noise. The colors are rich, and the lighting is sufficient. The texture details are not clear. The main subject clearly being the food in the middle. Therefore, the quality is poor.

Degradation-aware Text

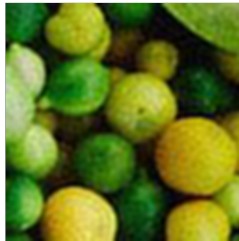 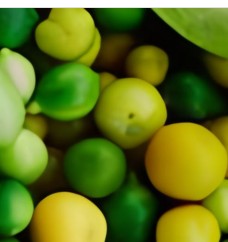 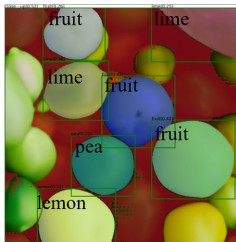

LR            DRM-processed Image            Semantic Mask + Tag +            Tags(DAPE)
                                              Bounding Box

avocado, citrus
fruit, fruit, green,
lemon, lime,
orange,
tangerine,

The image is a close-up of a collection of citrus fruits. The focus is precise, resulting in a clear image with no noticeable noise. The colors are rich, and the lighting is sufficient. The texture details are clear, and the composition is centered. However, the main subject is not clearly defined, making it difficult to distinguish the main content of the image. Therefore, the quality is poor.

Degradation-aware Text

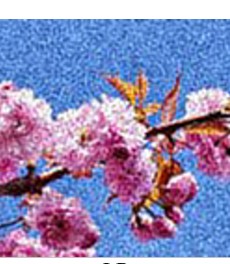 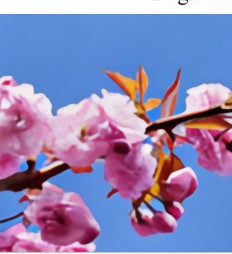 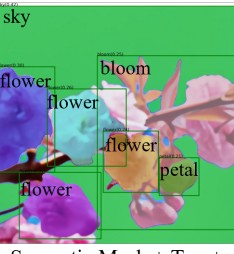

LR            DRM-processed Image            Semantic Mask + Tag +            Tags(DAPE)
                                              Bounding Box

bloom, blue,
branch, bud,
cherry blossom,
cherry tree, sky,
flower, tree,

The overall clarity of this image is acceptable. The main subject, the cherry blossom tree, retains most of its texture details. The background sky is somewhat blurry and there is some noise present. Therefore, the quality of this image is average.

Degradation-aware Text

Figure 6: Visualization of spatial-grounded textual and semantic-enhanced visual guidance, including objects with semantic masks and bounding boxes, tags generated by DAPE, and degradation-aware text. We align bounding boxes with object-level tags to get image fidelity guidance.

918
919
920
921
922
923
924
925
926
927
928
929
930
931
932
933
934
935
936
937
938
939
940
941
942
943
944
945
946
947
948
949
950
951
952
953
954
955
956
957
958
959
960
961
962
963
964
965
966
967
968
969
970
971

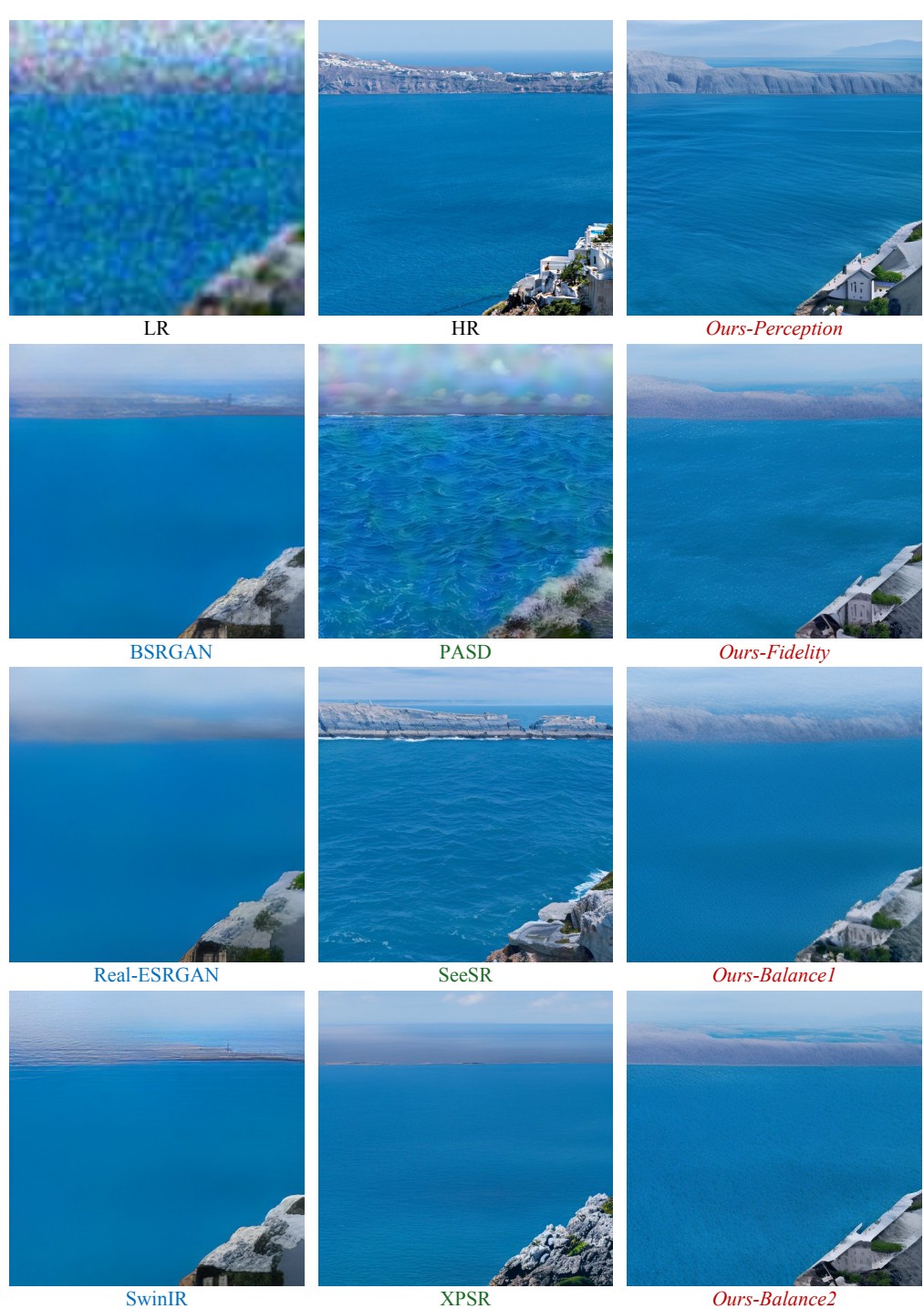

Figure 7: Qualitative comparisons on an example from DIV2K-Val. GAN-based methods tend to produce overly smooth outputs, making it difficult to recover clear images from those with complex noise. Diffusion-based methods are prone to hallucinations: SeeSR and XPSR mistakenly restore houses as rocks, while PASD interprets noise as water ripples. Our model and its variants preserve the realism of the seawater and maintain the general contours of the houses.

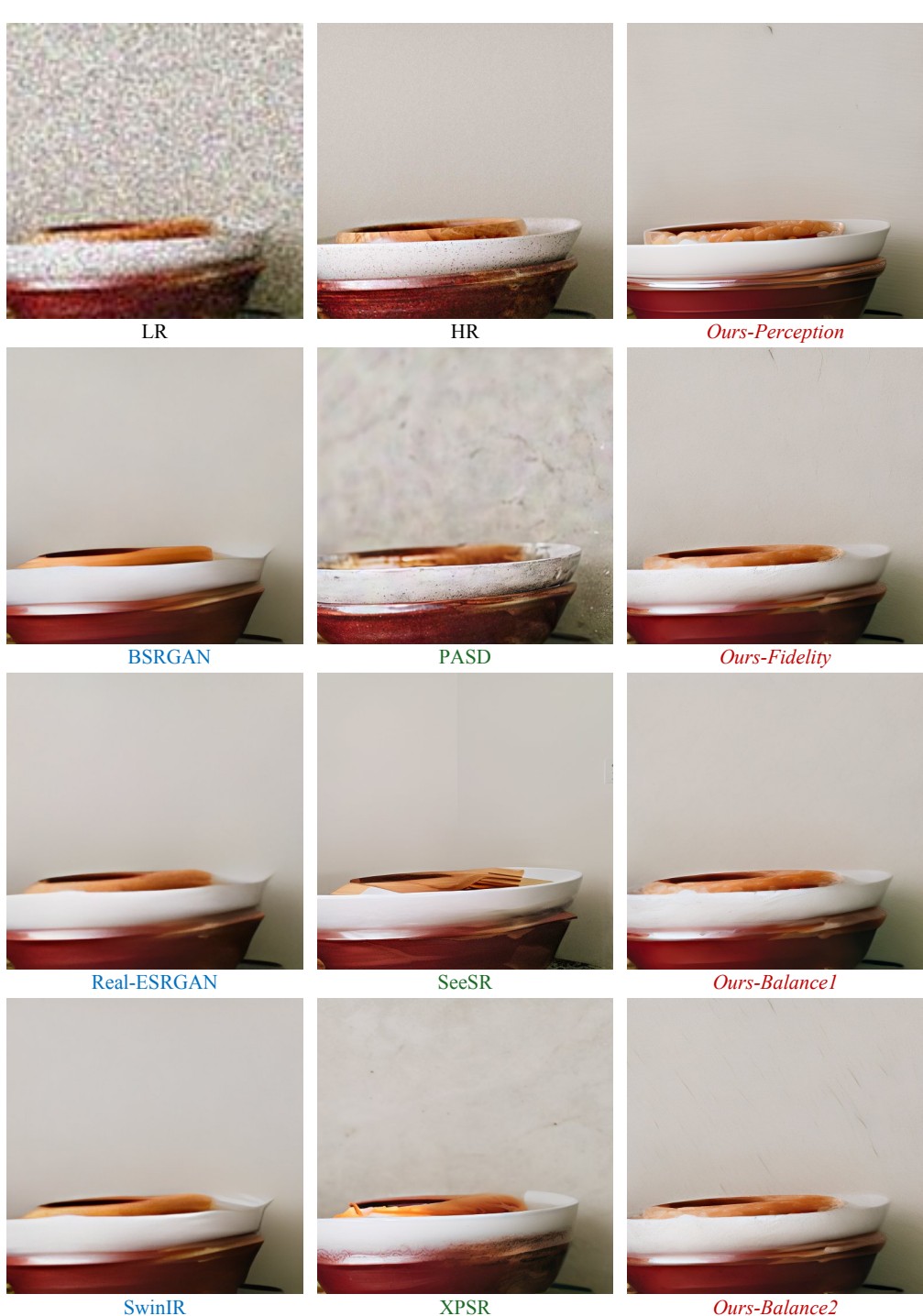

LR  HR  *Ours-Perception*

BSRGAN  PASD  *Ours-Fidelity*

Real-ESRGAN  SeeSR  *Ours-Balance1*

SwinIR  XPSR  *Ours-Balance2*

Figure 8: Qualitative comparisons on an example from DIV2K-Val. GAN-based methods lack detailed textures. Diffusion-based methods generate jade that is deformed. Our model and its variants generate richer texture details while maintaining fidelity, thereby more faithfully reflecting the jade and plate compared to diffusion-based methods.

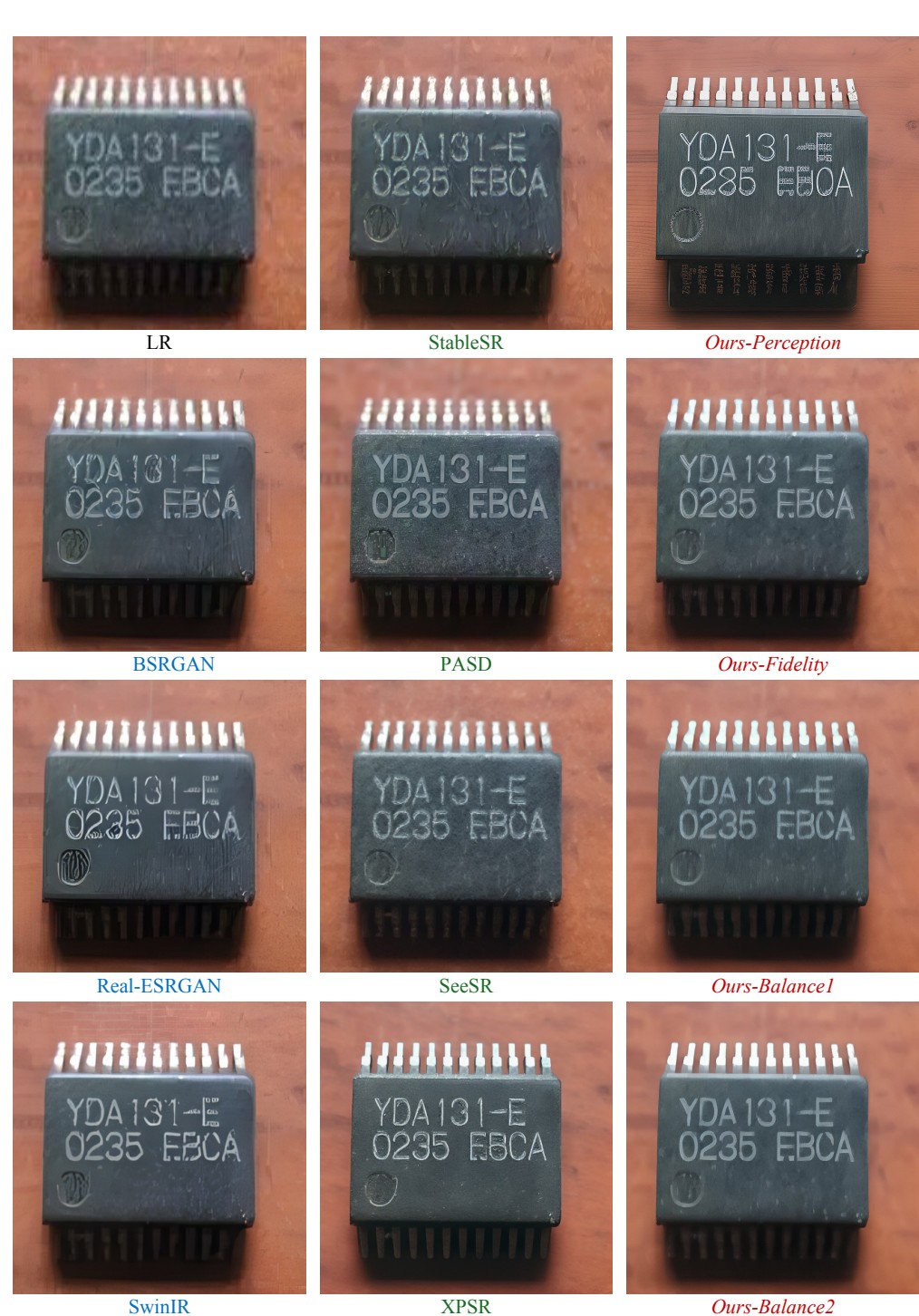

Figure 9: Qualitative comparisons on an example from RealLR200. GAN-based methods tend to oversmooth the chip surface and produce irregular font structures. Diffusion-based methods sharpen edges but introduce structural inconsistencies: StableSR and XPSR produce distorted or missing strokes in the characters, while PASD and SeeSR generate spurious textures on the chip body. Our model and its balanced variants restore the text more clearly and consistently, no spurious textures on the chip body, achieving a balanced trade-off between perceptual quality and distortion.

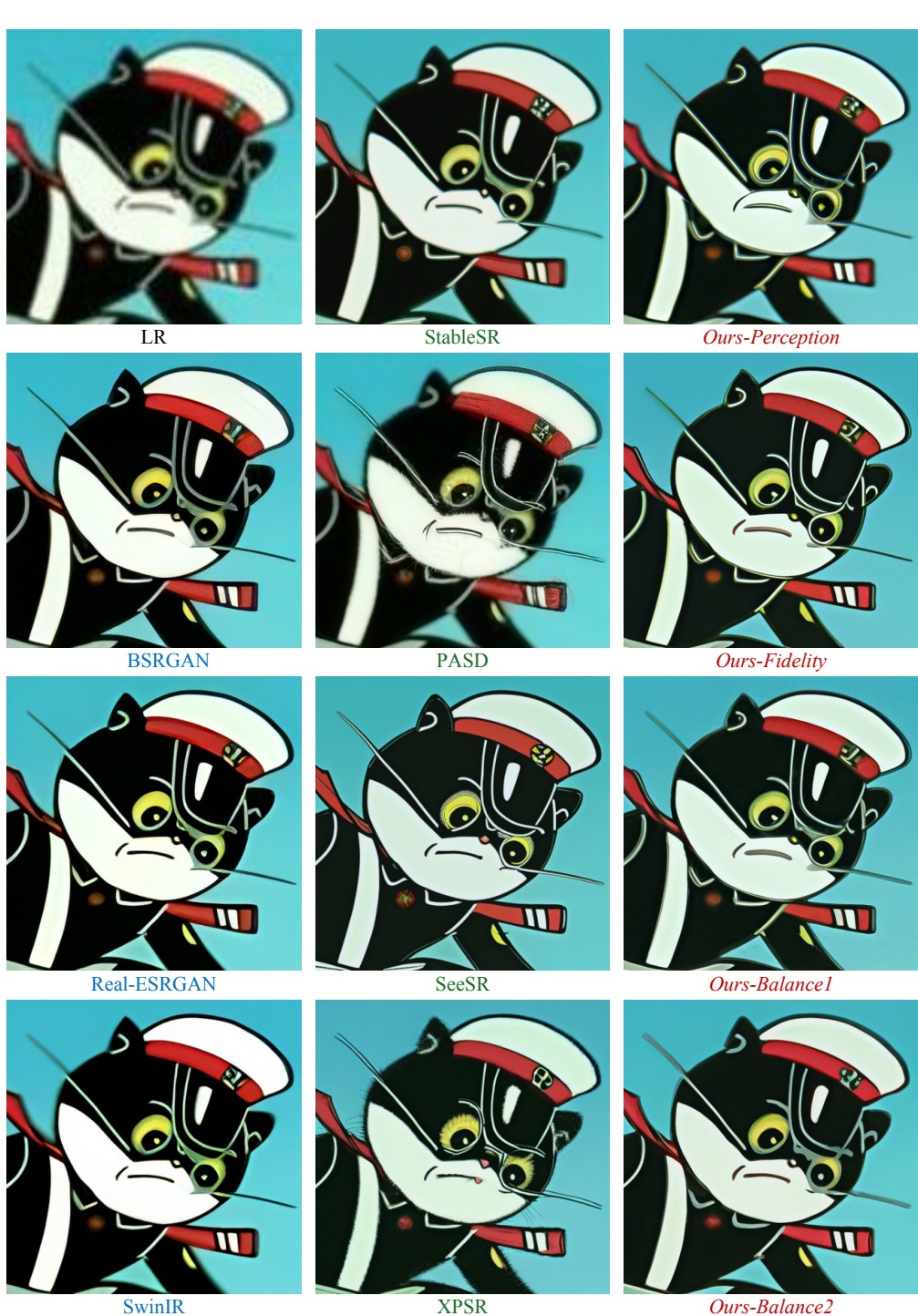

Figure 10: Qualitative comparisons on an example from RealLR200. While GAN-based methods remove noise effectively, they lead to blurred whiskers, linework, and edges. Diffusion-based methods yield sharper edges but are prone to hallucinations and geometric distortions: XPSR and PASD hallucinate cat fur, SeeSR produces abnormal eye patterns, and all four diffusion-based methods alter the cap badge. Our model and its balanced variants produce clean and stable linework without hallucinations, achieving a balanced trade-off between perceptual quality and distortion.

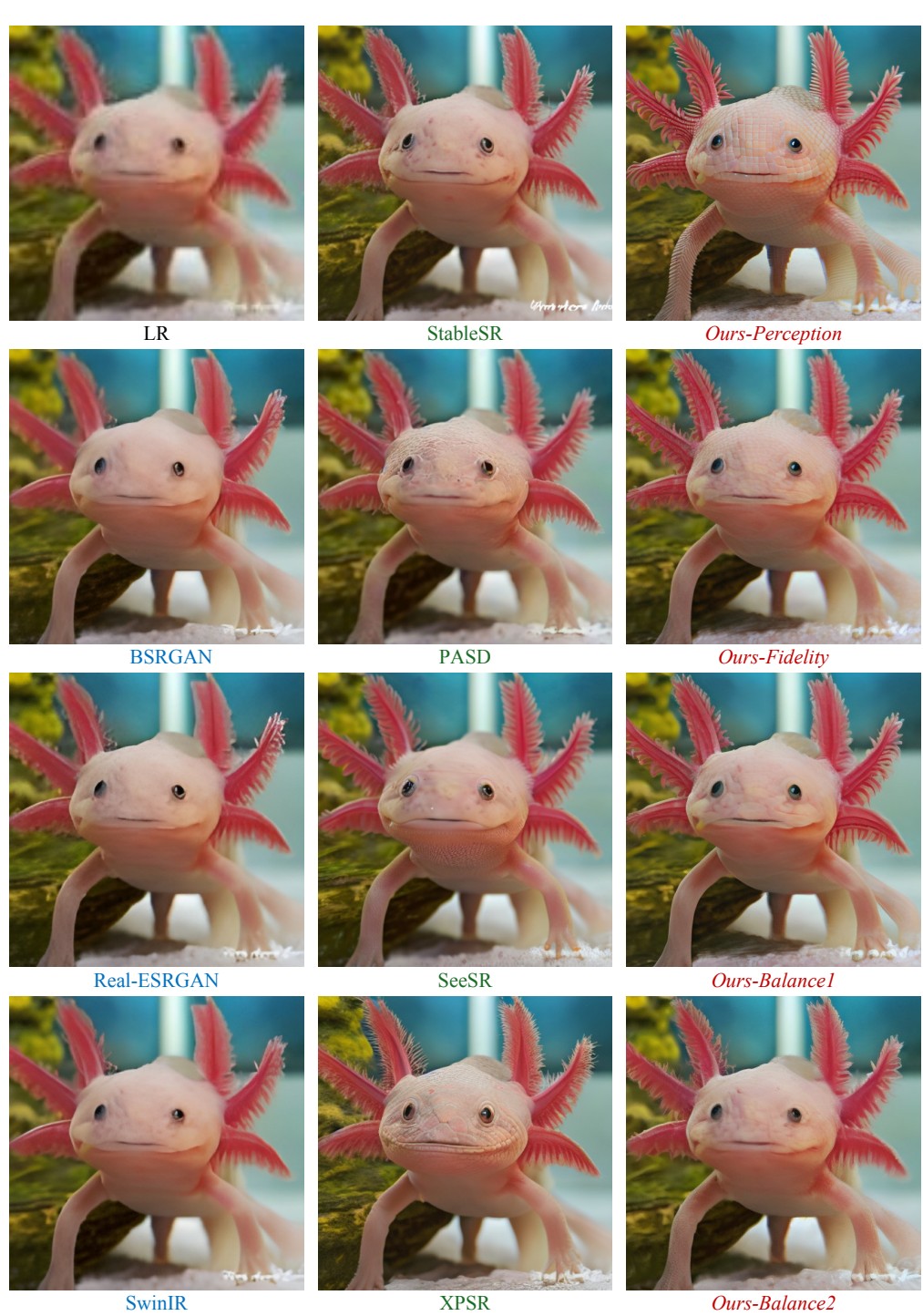

Figure 11: Qualitative comparisons on an example from RealLR200. GAN-based methods over-smooth the axolotl's gill filaments and skin, whereas diffusion-based methods generate more texture but introduce hallucinations and shape distortions: PASD, SeeSR, and XPSR hallucinate skin grain, SeeSR and XPSR warp the mouth contour, and StableSR injects blotchy textures. Our model and its balanced variants deliver clean, stable details, achieving a balanced trade-off between perceptual quality and distortion.

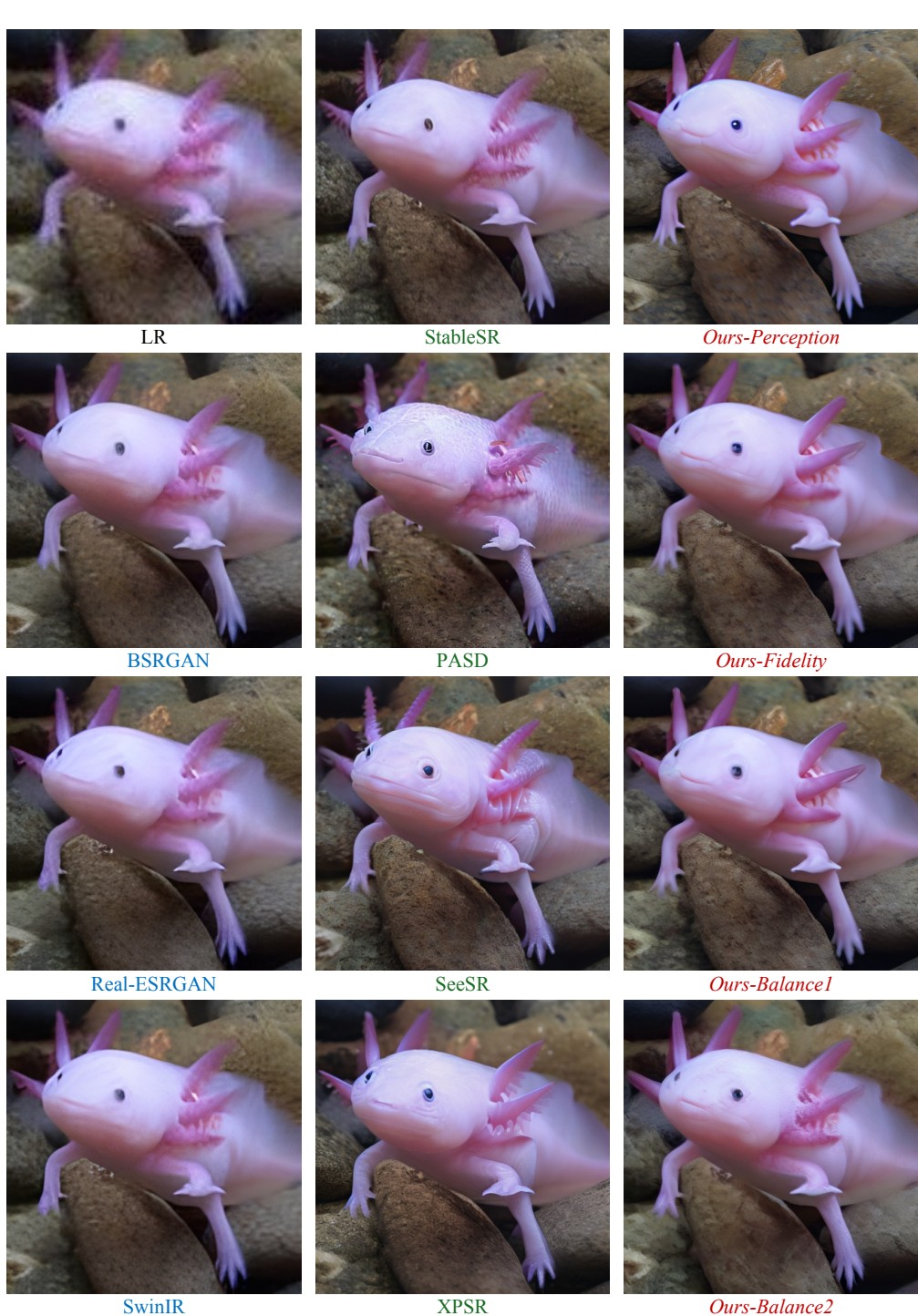

Figure 12: Qualitative comparisons on an example from RealLR200. GAN-based methods over-smooth the axolotl's gill filaments and body, blurring edges. Diffusion-based methods sharpen edges but introduce hallucinations and geometric distortions: PASD and SeeSR generate spurious skin textures and distorted mouths, while StableSR and XPSR hallucinate false filaments. Our model and its balanced variants keep natural skin and gill filaments, achieving a balanced trade-off between perceptual quality and distortion.

