# OpenReview forum: "Balancing Perception and Distortion in Super Resolution via Spatial–Semantic Guidance"
_ICLR.cc/2026/Conference — ICLR 2026 Conference Withdrawn Submission_

### Official Review · Reviewer_MDXU · 2025-10-29

**Soundness:** 2
**Presentation:** 2
**Contribution:** 2
**Rating:** 4
**Confidence:** 4

**Summary:**

This paper proposes a spatial–semantic guided SR framework, a diffusion-based approach aimed at pushing the perception–distortion boundary. First, a spatial-aware text fusion mechanism is introduced to improve fidelity. Two-branch image encoders are employed to extract low-level and high-level features for perception enhancement. Four variants of the proposed method are designed to validate the effectiveness of different contributions. Comparisons are made with both GAN-based and diffusion-based methods.

**Strengths:**

The starting point—addressing the trade-off between perception and distortion—is good. Most current works focus on improving the generative capability of diffusion-based methods and perform poorly on fidelity. The authors recognize this issue and attempt to improve it.

**Weaknesses:**

1. The method does not effectively improve the perception–distortion trade-off. Comparisons rely on different variants rather than a single model performing well on both fidelity and perception metrics.

2. The architecture is overly complex, with many modules and cumbersome naming, which hurts readability. Some components (spatial-aware text attention, semantic-enhanced image attention) lack concrete procedural descriptions.

3. The ControlNet internal attention design appears ad hoc and is not theoretically or empirically justified.

4. The role of each module is not clearly substantiated. Why does spatial-grounded textual guidance contribute to fidelity? From Table 3, the image encoder contributes the most to fidelity, yet the authors claim that the image and semantic encoders are intended to improve perception. Table 3 indicates that only the semantic encoder improves perception.

**Questions:**

1. Can the authors demonstrate a single model (not different variants) that achieves strong performance on both fidelity and perception metrics?

2. What are the precise implementations for spatial-aware text attention and semantic-enhanced image attention?

3. What is the rationale behind the ControlNet internal attention design, and can the authors provide ablations or analysis to justify it?

4. Why should spatial-grounded textual guidance contribute to fidelity? Please provide evidence or analysis supporting this claim.

---

### Official Review · Reviewer_dJJ8 · 2025-10-31

**Soundness:** 3
**Presentation:** 3
**Contribution:** 2
**Rating:** 2
**Confidence:** 5

**Summary:**

This paper addresses the critical perception-distortion trade-off in image super-resolution. The authors observe that GAN-based methods suffer from blurry textures despite high fidelity, while diffusion-based models produce perceptually rich but often hallucinatory results. To resolve this, the paper proposes SpaSemSR, a framework that introduces two complementary guidance mechanisms to steer the diffusion process. The main contributions are:
1. Spatial-grounded textual guidance: Embeds spatial location information into text prompts to ensure semantic guidance is spatially consistent, thereby reducing structural distortion.
2. Semantic-enhanced visual guidance: Employs a dual-branch image encoder (for structure and semantics) and a semantic degradation constraint to improve semantic understanding of low-quality inputs.
3. A novel fusion mechanism: A Spatial-Semantic ControlNet with dedicated attention layers fuses these cross-modal guidances to suppress hallucinations and preserve fidelity during diffusion.

**Strengths:**

1. The work is validated using a strong set of metrics, including both reference-based fidelity measures (PSNR, SSIM) and non-reference perceptual scores (CLIP-IQA, MUSIQ, MANIQA), which is essential for assessing the perception-distortion trade-off.
2. The proposed method is extensively compared against leading GAN-based and diffusion-based approaches, effectively demonstrating its ability to strike a better balance.
3. The paper is well-written and logically organized. The motivation is clearly articulated, and the high-quality figures (e.g., Fig. 1 and Fig. 2) are highly effective at illustrating the problem and the proposed solution.

**Weaknesses:**

1. The framework is highly complex, integrating multiple large pre-trained models. The lack of discussion on inference speed and memory usage makes it difficult to assess its practical viability.
2. The method's performance relies heavily on the accuracy of Grounded-SAM, which can be brittle on severely degraded inputs. The reliance on a pre-processing Degradation Removal Model (DRM) adds computational overhead and risks error propagation.
3. The paper describes SpaTextAtten and SemImgAtten as novel layers. However, they appear to be standard cross-attention mechanisms. Their novelty seems to lie in the application to the proposed guidance signals rather than the architectural design, which could be an overstatement of contribution.

**Questions:**

1. How does the system perform when Grounded-SAM fails (e.g., on non-rigid objects, severe occlusions, or extreme degradations)? What are the failure cases of the DRM pre-processing step, and does it risk introducing new artifacts that could mislead the guidance?
2. Does this approach require training and deploying four separate models to cater to different user preferences? Have you explored a more flexible method to navigate the trade-off with a single model at inference time, perhaps via a tunable parameter?
3. Could you please clarify whether the novelty of SpaTextAtten and SemImgAtten lies in their internal architecture or in their application of standard cross-attention to your novel guidance signals?
4. Could you provide quantitative data on inference speed (e.g., time to generate a 512x512 image) and GPU memory requirements, and compare them against key baselines like Real-ESRGAN and StableSR?
5. What was the motivation for using SDv2 as the base model? Have you considered or experimented with more recent backbones like SDXL or SD3 to see if they further improve performance?
6. According to Table 2, the Balance-1 and Balance-2 variants do not achieve top performance on non-reference perceptual metrics compared to other diffusion-based methods. This seems to contradict the goal of achieving a superior balance. Could you comment on this?
7. The dedicated variants do not seem to consistently demonstrate their intended strengths. For instance, the Fidelity variant often appears less sharp than GAN models, and the Perception variant does not always show a clear perceptual advantage over other diffusion methods. Could you please explain this apparent discrepancy between the variants' stated goals and their empirical performance?

---

### Official Review · Reviewer_XNj8 · 2025-10-31

**Soundness:** 3
**Presentation:** 3
**Contribution:** 2
**Rating:** 4
**Confidence:** 4

**Summary:**

This work proposes SpaSemSR, which aims to effectively tackle the perception-distortion trade-off. Specifically, the primary aim is to introduce diffusion based generative priors, without unnecessary fidelity-loss. Accordingly, SpaSemSR introduces 1) spatial-grounded textual guidance and 2) semantic-enhanced visual guidance to achieve this. Experimental results verify that these external priors can be effectively integrated by a spatial-semantic attention, thereby improve the overall performance in terms of perception-distortion trade-off.

**Strengths:**

- The overall writing is very clear and easy to follow.
- The complementary aspect of the proposed "spatial-grounded textual guidance" and "semantic-enhanced visual guidance" is interesting.
- The proposed (technical) methods to adapt each of above into the SR pipeline is technically sound.
- Visual results from the proposed method is impressive and often outperforms baseline methods.

**Weaknesses:**

The overall method is clear, sound and interesting. However, at the current state, I will set my position to border reject. My primary concerns are about the overall performance of SpaSemSR, since it is hard to evaluate it due to 1) the presentation manner in the current state and 2) missing baselines.

**I am willing to update my score to accept** if the authors can sufficiently address the concerns regarding the performance specified as below.

---

**Weakness 1: Overall Performance**

Despite SpaSemSR uses more external large-scale pretrained models, XPSR often outperforms SpaSemSR in terms of perceptual metrics (CLIP-IQA, MUSIQ, MANIQA). While SpaSemSR achieves higher distortion metrics (PSNR, SSIM), the trade-off relationship is hard to compare with single numbers.

---

**Weakness 2: Perception-Distortion (PD) Trade-off**

Related to Weakness1, while the paper focuses on the PD trade-off, the current presentation makes it hard to analyze the PD trade-off. (Thus, it is hard to determine whether SpaSemSR really outperforms XPSR).

Please provide PSNR-to-CLIPIQA / PSNR-to-MUSIQ / PSNR-to-MANIQA trade-off curve graphs for each dataset (total 9 graphs).
That is, for each graph, I would expect 1) a curve that interpolates **Ours-perceptual - Ours-balance - Ours-fidelity**, and 2) points indicating scores for each baseline (including XPSR).
Examples are as **Fig.17 of SRFlow [5]** and **Fig.4 of AESOP [6]** (More examples are in the supplementary materials of each).

[5] (SRFlow) Lugmayr, Andreas, et al. "Srflow: Learning the super-resolution space with normalizing flow." European conference on computer vision. Cham: Springer International Publishing, 2020.

[6] (AESOP) Lee, MinKyu, et al. "Auto-Encoded Supervision for Perceptual Image Super-Resolution." Proceedings of the Computer Vision and Pattern Recognition Conference. 2025.

---

**Weakness 3: Additional Baselines 1**

> Thus, neither paradigm fully resolves the tension between perception and distortion, leaving it as an unsolved issue. (Line 53)

> Existing balancing techniques are mostly GAN-centric. (Line 134)

Regarding the words from the authors above, the baseline works cited by the authors indeed do not primarily focus on the perception-distortion trade-off. However, there has been recent Diffusion based works to tackle this issue as PiSA-SR [1] (PD trade-off interpolation)  and FaithDiff [2] (Distortion aware). Please further discuss and compare the performance with these; or counter-argue if these are not valid methods to compare.

[1] (PiSA-SR) Sun, Lingchen, et al. "Pixel-level and semantic-level adjustable super-resolution: A dual-lora approach." Proceedings of the Computer Vision and Pattern Recognition Conference. 2025.

[2] (FaithDiff) Chen, Junyang, Jinshan Pan, and Jiangxin Dong. "Faithdiff: Unleashing diffusion priors for faithful image super-resolution." Proceedings of the Computer Vision and Pattern Recognition Conference. 2025.

---
**Weakness 4: Additional Baselines 2**

SpaSemSR utilizes many external priors. There has been recent works that utilizes external priors similarly. Please further discuss and compare the performance with works as MMSR [3] and SUPIR [4]; or counter-argue if these are not valid methods to compare.

[3] (MMSR) Mei, Kangfu, et al. "The power of context: How multimodality improves image super-resolution." Proceedings of the Computer Vision and Pattern Recognition Conference. 2025.

[4] (SUPIR) Yu, Fanghua, et al. "Scaling up to excellence: Practicing model scaling for photo-realistic image restoration in the wild." Proceedings of the IEEE/CVF conference on computer vision and pattern recognition. 2024.

**Questions:**

**Question 1**

SpaSemSR utilizes a ControlNet-style adpater with external large-scale pretrained models. Since these often require heavy computational cost, please specify the overall and per-element computational cost (Num Params, Gflops, Wall clock time, etc), and compare it against baselines.

---

### Official Review · Reviewer_ripw · 2025-10-31

**Soundness:** 1
**Presentation:** 2
**Contribution:** 1
**Rating:** 2
**Confidence:** 4

**Summary:**

The authors aim to address the fidelity-perception trade-off in diffusion-based image super-resolution. Specifically, they propose degradation-aware and spatial-aware mechanisms to assist in the image SR process.

**Strengths:**

The authors provide extensive details to explain the methodology and offer multiple visual examples to demonstrate the effectiveness of their approach.

**Weaknesses:**

(1)	The first figure lacks strong persuasiveness. When the degradation of the LR image is so severe that even the human eyes cannot predict the corresponding details, discussing the fidelity of the results becomes ambiguous. Is the fidelity and hallucination discussed compared to the LR or the HR image? This is a difficult issue to clarify. It is recommended to include a more intuitive figure in Figure 1, where it is visually evident that the model's results exhibit significantly fewer hallucinations and higher consistency with respect to both the LR and HR images.

(2)	In Section 3.2, the authors argue that the lack of fidelity in current T2I-generated images is due to the limited spatial understanding of the given text conditions, leading to misalignment between the semantics in the text prompt and the actual image semantics. As the core motivation of the paper, this claim lacks strong supporting evidence.

(3)	The comparison with other methods is limited. There are recent works, such as PiSA-SR [1], that also discuss the fidelity-perception trade-off. The authors should provide a more comprehensive comparison, including the latest works that consider this trade-off.

(4)	The innovation is limited. The two core mechanisms proposed by the authors are spatially-aware representations (relying on Grounded-SAM) and degradation-aware representations. Similar designs for these mechanisms have already been explored in numerous existing works.

(5)	The article is challenging to understand, as the complex and fancy words are often used to depict an easy operation. Simpler language sometimes could be much clearer.

[1] Sun L, Wu R, Ma Z, et al. Pixel-level and semantic-level adjustable super-resolution: A dual-lora approach[C]//Proceedings of the Computer Vision and Pattern Recognition Conference. 2025: 2333-2343.

**Questions:**

(1)	In the authors' motivation, a key issue is whether the performance limitations are indeed caused by inappropriate text prompts. If the prompts could better align with the image (including spatial layout and degradation level), would this lead to results with higher fidelity? If this inference holds, would the relationship between perception and fidelity still be contradictory in such a scenario?

(2)	Another critical question is whether the design of text prompts should consider not only factors related to the image itself but also the distribution of the T2I model. For instance, as proposed in works like Vivid-VR [2], is it possible that the text extracted from the LR image and the text prompts familiar to the pretrained T2I model are from different distributions, and that this also affects the fidelity of the generated images?


[2] Bai H, Chen X, Yang C, et al. Vivid-VR: Distilling Concepts from Text-to-Video Diffusion Transformer for Photorealistic Video Restoration[J]. arXiv preprint arXiv:2508.14483, 2025.

---

### Note · Authors · 2025-11-13

I have read and agree with the venue's withdrawal policy on behalf of myself and my co-authors.